# The angiosperm radiation played a dual role in the diversification of insects and insect pollinators

**David Peris** [1] ✉ **& Fabien L. Condamine** [2]

Interactions with angiosperms have been hypothesised to play a crucial role in driving diversification among insects, with a particular emphasis on pollinator insects. However, support for coevolutionary diversification in insect–plant interactions is weak. Macroevolutionary studies of insect and plant diversities support the hypothesis that angiosperms diversified after a peak in insect diversity in the Early Cretaceous. Here, we used the family-level fossil record of insects as a whole, and insect pollinator families in particular, to estimate diversification rates and the role of angiosperms on insect macroevolutionary history using a Bayesian process-based approach. We found that angiosperms played a dual role that changed through time, mitigating insect extinction in the Cretaceous and promoting insect origination in the Cenozoic, which is also recovered for insect pollinator families only. Although insects pollinated gymnosperms before the angiosperm radiation, a radiation of new pollinator lineages began as angiosperm lineages increased, particularly significant after 50 Ma. We also found that global temperature, increases in insect diversity, and spore plants were strongly correlated with origination and extinction rates, suggesting that multiple drivers influenced insect diversification and arguing for the investigation of different explanatory variables in further studies.

Today, flowering plants (angiosperms) dominate most terrestrial ecosystems and provide an important part of the food chain and niche requirements for many other organisms. It is estimated that angiosperms account for about 90% of all land plants (embryophytes), or about 300,000 living species[1]. Their origin was one of the most transformative events in Earth's history. However, the age of crown angiosperms remains highly uncertain[2]. There is almost universal molecular support for a pre-Cretaceous origin of stem angiosperms[3], ranging from 310 to 380 million years ago (Ma) depending on the study[4–7]. In contrast, the earliest fossil remains that can be assigned with high confidence to crown angiosperms are tricolpate pollen grains from the Barremian–Aptian transition (121 Ma[8]). The fossil record still provides fundamental evidence for the origin of

angiosperms around 250–140 Ma and their explosive radiation since the Cretaceous[9–11]. Despite the uncertainty in the timing of the origin of angiosperms, it seems clear that the diversification of the major angiosperm lineages occurred during the Angiosperm Radiation (125–90 Ma, e.g.[12]), the Cretaceous Terrestrial Revolution (125–80 Ma[13]), or in the Cretaceous and Cenozoic, in the recently termed Angiosperm Terrestrial Revolution (ATR, 100–50 Ma[10]) (Fig. 1). During this interval, conifers and other plant lineages substantially declined in diversity[14,15].

By contrast, the major radiation of modern insect lineages began around 245 Ma[16,17], apparently long before the radiation of angiosperms. Since the Jurassic, insect families have shown low extinction rates[16,18,19]. Insect family richness peaked transiently during the Early

[1]Institut Botànic de Barcelona (CSIC-CMCNB), 08038 Barcelona, Spain. [2]CNRS, Institut des Sciences de l'Evolution de Montpellier, Université de Montpellier, Place Eugène Bataillon, 34095 Montpellier, France. ✉e-mail: david.peris@ibb.csic.es

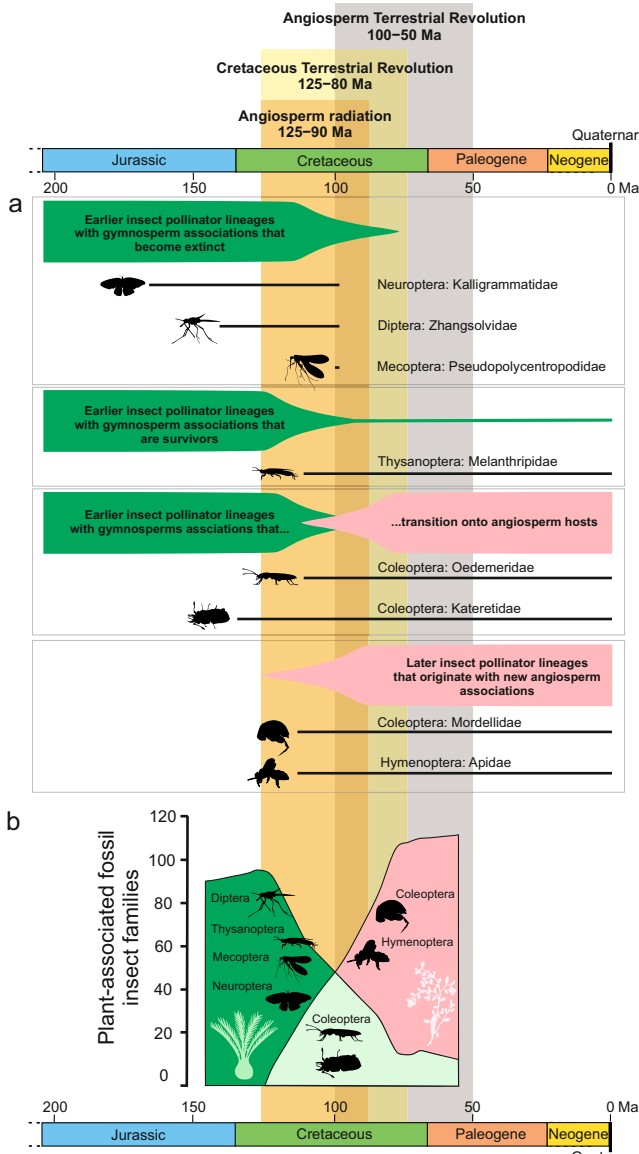

**Fig. 1 | Different pollination modes showing different groups of insects under distinctive patterns of extinction, survival, and origination following the gymnosperm-angiosperm transition.** The fossil pollination cases reviewed in refs. 28,38. The periods of Angiosperm Radiation[12], Cretaceous Terrestrial Revolution[13], and Angiosperm Terrestrial Revolution[10] are marked. **A** Diverse fossil community of Cretaceous pollinators and the lifespan of these families. **B** Representation of the transition from gymnosperm-associated insect families to angiosperms-associated insect families, illustrating examples of known pollinators groups in the different transitional situations. Data extracted from ref. 35.

Cretaceous around 125 Ma, when angiosperms were still rare[20,21]. This peak in insect richness occurred before extinctions within early-diverging groups, and is partly related to the mid-Cretaceous floral turnover accompanying the evolution of flowering plants[14,20] (Fig. 1). Therefore, the Early Cretaceous richness peak of insects may reflect a transitional period in insect evolution during which radiating extant families coexisted with early-diverging ones that are now rare or have gone extinct[20,22–24]. However, this transient peak is also partly correlated with the extremely fossiliferous deposits of the time, such as the Yixian Formation in China, the Crato Formation in Brazil, or the Myanmar amber[20]. In any case, the fossil record and phylogenetic studies support a scenario in which all extant and extinct orders of insects, including angiosperm pollinators, evolved well before the Early Cretaceous with origins that largely predate the diversification of

crown angiosperms, except for the lineages of some flies, bees, and long-proboscid butterflies[25].

It was thought that the first seed plants were wind-pollinated until some insects diversified and began to feed on gymnosperm ovule secretions in a surface-fluid-feeding manner or on gymnosperm pollen[26,27]. However, we now know that Cretaceous plants were pollinated by a different spectrum of pollinator agents[28], predating that of nectar- and pollen-feeding insects on angiosperms[29] (Fig. 1). The complex interactions between potential pollinators and gymnosperms appear to have persisted at least since the early Permian (283–273 Ma), predating the first flowering plants by more than 100 Ma[30,31]. Since then, there is a diverse, well-documented Cretaceous pollinator community found in sediments and ambers, supporting gymnosperm–insect pollination modes and host associations with ginkgoaleans, cycads, conifers, and bennettitalean gymnosperms during the Early Cretaceous and the early Late Cretaceous[27,28,32–36] (Fig. 1). In contrast, evidence for angiosperm-pollen consumers or flower visitors does not appear in the fossil record until the Late Cretaceous, around 99 Ma[28,36,37]. Insects continued pollinating flowering plants since then[38,39]. The ensuing question is: what effect did the radiation of angiosperms since the Cretaceous have on the diversification of insects, and in particular pollinating insects, given that insects have been diversifying with and pollinating gymnosperms since the Permian?

Elucidating when, how and why the ecological transformation of ecosystems began with the coevolution between insects and angiosperms has become a topic of recent interest[10,25,39,40]. Relying on qualitative comparisons has limitations, and there are few quantitative tests to assess whether coevolution has driven diversification between plants and insects. In this study we tested the widely held hypothesis of the impact of angiosperm radiation on insect diversification in general, and on insect pollinators in particular, using the fossil record. To complement our results, we also analysed the role that five additional variables (diversity dependence, gymnosperms, spore plants, continental fragmentation, and temperature) may have had on insect diversification rates. We hypothesise that the radiation of angiosperms had a positive effect on insect diversification rates, mainly by increasing origination rates, but may also have reduced extinction rates by facilitating the shift of gymnosperm pollinators to angiosperms as angiosperms diversified and gymnosperms entered a phase of diversity decline from the Late Cretaceous onwards.

## Results

To estimate whether insect origination and/or extinction rates varied through time and whether they could be correlated to the angiosperm radiation, we retrieved the family-level fossil dataset of ref. 19. We used the Bayesian process-based approach implemented in PyRate 3[41], which models simultaneously the rates of preservation and their variation across taxa, the times of origination and extinction of each taxon, and then the rates of origination and extinction through time. By estimating the preservation rates, PyRate corrects the ages of origination and extinction of each taxon that can alleviate known issues of the fossil record such as the temporal incompleteness of the fossil record[42]. As expected, we recovered a dynamic accumulation of family diversity with major increases in family richness during the Cretaceous and the Cenozoic (Fig. 2a), in line with previous studies[20]. We also repeated the analysis, including only the families from this same fossil dataset that were cited as extant or extinct pollinator[28], including also the possible pollinators to be as inclusive as possible (Fig. 2a). To clearly differentiate a fossil pollinator and a fossil possible pollinator, we relied on a recent review of insect pollination through deep time[28].

We then relied on the Bayesian multivariate birth-death (MBD) model implemented in PyRate to simultaneously estimate correlations between diversification dynamics and multiple environmental variables, with the statistical support being estimated with a shrinkage

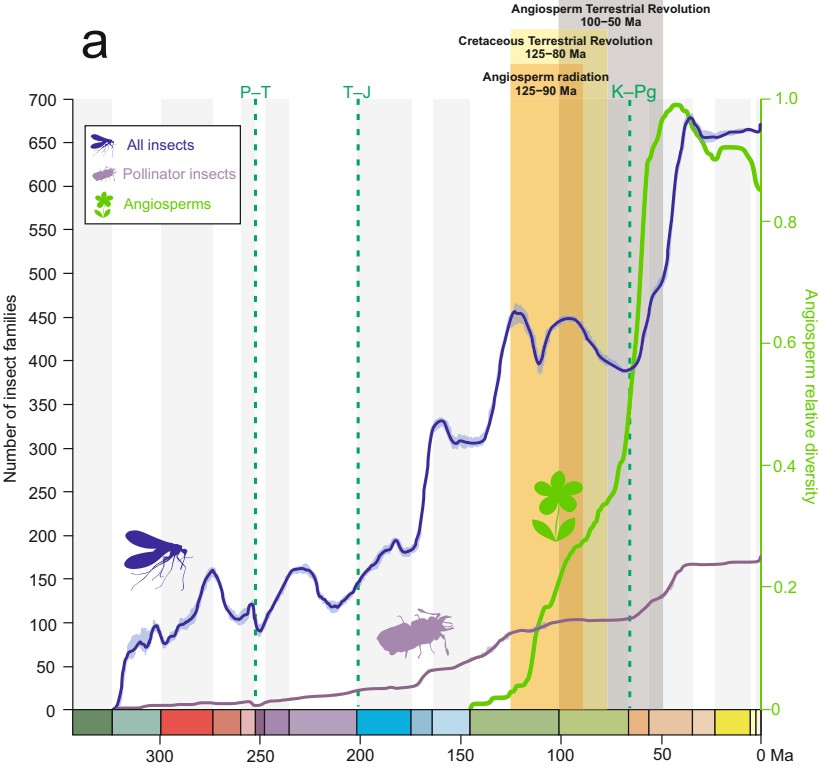

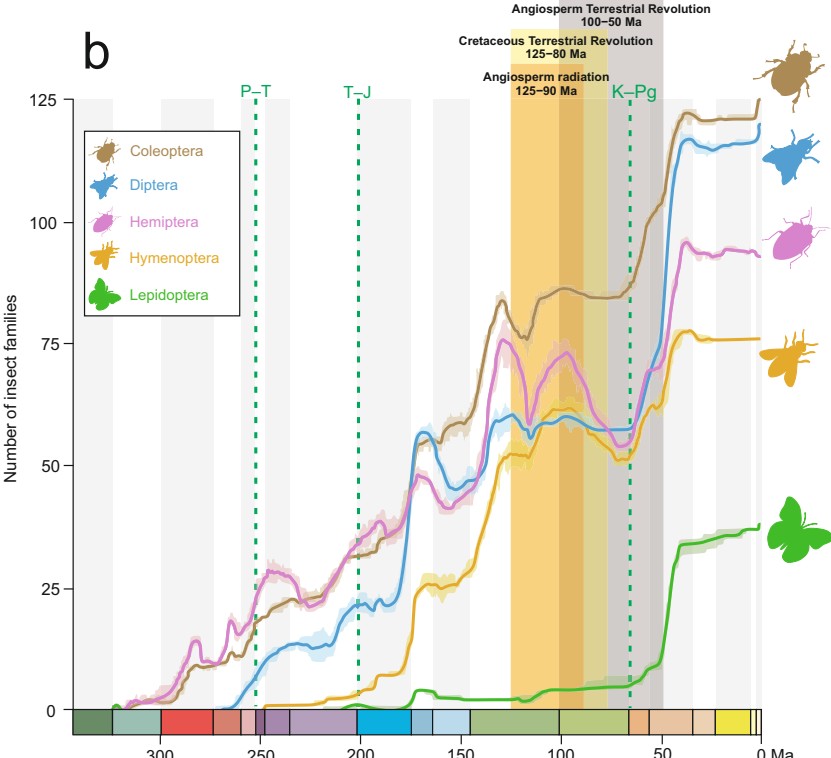

**Fig. 2 | Accumulated diversification of insect families through geological time estimated using the software PyRate and the family-level fossil dataset of ref. 19 compared with the angiosperm relative diversity.** Dark green dotted lines indicate the Permian–Triassic (P–T), Triassic–Jurassic (T–J), and the Cretaceous–Paleogene (K–Pg) mass extinctions. The periods of Angiosperm Radiation[12], Cretaceous Terrestrial Revolution[13], and Angiosperm Terrestrial Revolution[10] are marked. **A** All insect diversity, specifically the pollinator insect diversity, in the left axis; the angiosperm relative diversity from ref. 6, put in shape in ref. 14, in the right axis. **B** Family accumulation through time for five selected insect orders.

weight ($\omega$) for each correlation parameter ($G$) for origination ($G\lambda$) and extinction ($G\mu$) depending on each environmental variable[43]. We focused on the role of angiosperms but are aware that several other drivers may have also complementary impacted the diversification of insects (see Methods for details). In addition to the relative diversity of angiosperms, we thus incorporated five additional variables: the number of insect families through time (diversity dependence), diversity of gymnosperms, and spore plants[6], the continental fragmentation[44], and global temperature[45].

We first estimated origination and extinction rates over the entire evolutionary history of insects (i.e. from the mid-Carboniferous to the present), for which angiosperms were absent from the fossil record until the Early Cretaceous. Our modelling results show that, of the six tested variables, angiosperms could have promoted a faster diversification of insects once they co-diversified since the Cretaceous (Supplementary Data 1a). Specifically, we found that the rise of flowering plants not only could have driven the origination of insect families ($\omega\lambda = 0.576$ and $G\lambda = 0.823$), but they could have also strongly buffered them against extinction ($\omega\mu = 0.802$ and $G\mu = -1.799$; Supplementary Data 1a). We then performed the same diversification analyses but with rates only estimated for the time interval covering the ATR (*sensu* ref. 10, from 100 to 50 Ma) and another set of analyses with rates only estimated in the aftermath of the ATR (from 50 Ma to present). The ATR-centred analyses indicated that this period is strongly correlated with a reduced insect extinction rates ($\omega\mu = 0.95$ and $G\mu_{ATR} = -4.555$) but did not affect insect origination rates (Fig. 3, Supplementary Data 1b). The post-ATR analyses indicated that this period is strongly correlated with an increased insect origination rates ($\omega\lambda = 0.645$ and $G\lambda_{post-ATR} = 1.052$) and decreased insect extinction rates ($\omega\mu = 0.797$ and $G\mu_{post-ATR} = -1.791$), but lesser than during the ATR (Fig. 3, Supplementary Data 1c). Therefore, angiosperms seem to have a dual role that has changed through time with an attenuation of the extinction in the Cretaceous and the beginning of the Cenozoic and a driver of origination in the Cenozoic, from 50 Ma onwards.

Second, we tested whether there was also a relationship between angiosperm radiation and the pollinating insect families by repeating the MBD analysis to estimate origination and extinction rates across pollinator groups only (including the possible pollinators, according to ref. 28). Our Bayesian modelling results shows that, among the six tested variables, angiosperms may have promoted diversification of

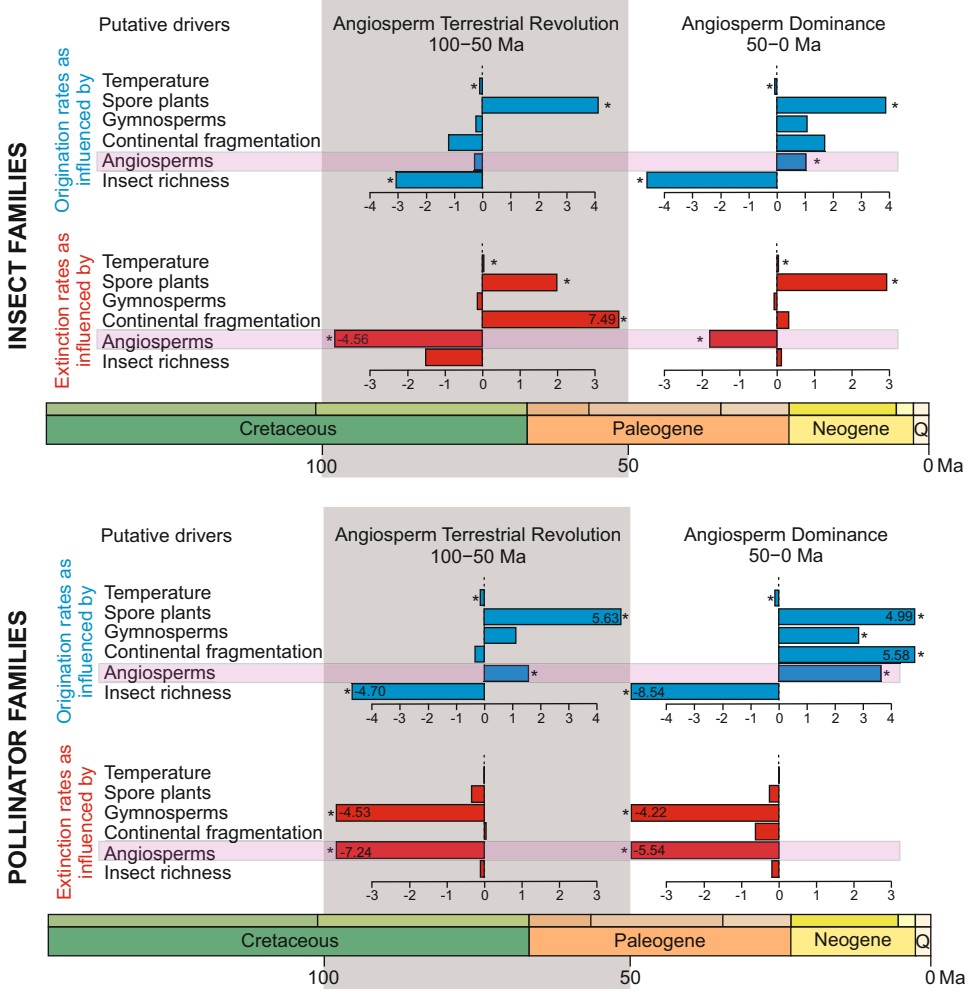

**Fig. 3 | Correlation trends of different analysed drivers for origination (in blue) and extinction (in red) rates on insect diversity for two periods of time: the Angiosperm Terrestrial Revolution timeframe (100–50 Ma[10]), and for the Angiosperm Dominance period (50–0 Ma).** The results for the same drivers analysing only the pollinator insect families, including the possible pollinator lineages. Data used in this representation represent median estimates and the 95% CI are presented in Supplementary Data 1–2. Drivers from top to bottom are Global mean temperature (Temperature), Spore plant relative diversity (Spore plants), Gymnosperm relative diversity (Gymnosperms), Continental fragmentation, Angiosperm relative diversity (Angiosperms), and Insect family richness (Insect richness). Asterisks indicate significant correlations recovered with the MBD model (shrinkage weight >0.5 and 95% CI not overlapping with zero). If any of the dates is out of their corresponding scale, it represents their value inside the box. The x-axes have no unit scale for correlation parameters in the MBD analyses.

pollinator insects once they had co-diversified since the Cretaceous not only due to a strong and positive correlation with origination rates ($\omega\lambda = 0.943$ and $G\lambda = 3.663$) but also to a strong and negative correlation with extinction rates ($\omega\mu = 0.967$ and $G\mu = -5.487$) (Supplementary Data 2a). Thus, the global rise of flowering plants may not only have driven the emergence of insect pollinator families but may also have buffered them against extinction, as similarly found for the whole insect record. Performing the same diversification analyses with rates estimated only for the time interval covering the ATR (from 100 to 50 Ma) and another set of analyses with rates estimated only in the post-ATR period (from 50 Ma to the present), we found that the ATR-centred analyses indicated that the rise of angiosperms was strongly correlated with a decrease in pollinator extinction rates ($\omega\mu = 0.978$ and $G\mu_{ATR} = -7.239$) and also with an increase in pollinator origination rates ($\omega\lambda = 0.799$ and $G\lambda_{ATR} = 1.558$; Fig. 3, Supplementary Data 2b). This evolutionary pattern is maintained in the Cenozoic, with the post-ATR analyses showing that this period is strongly correlated with both an increase in insect origination rates and a decrease in insect extinction rates. Compared to rates during the ATR, the correlation with origination rates is stronger ($G\lambda_{ATR} = 1.558$ vs. $G\lambda_{post-ATR} = 3.638$), while the correlation with extinction rates is weaker ($G\mu_{ATR} = -7.239$ vs. $G\mu_{post-ATR} = -5.535$; Fig. 3, Supplementary Data 2c).

With the aim of accounting for any bias in our results due to heterogeneity in preservation rates for specific groups in the fossil

record[46,47], we repeated the MBD analyses to test whether the relationship between angiosperm radiation and the five most diverse insect orders (i.e. Coleoptera, Diptera, Hemiptera, Hymenoptera, and Lepidoptera) held. We collected specific families by orders and found idiosyncratic family-level accumulation curves, suggesting different diversification dynamics (Fig. 2b), which is expected given their differences in life history, key innovations and ages (e.g. [19,46–49]). By dividing the timeframe into the ATR (100–50 Ma) and the post-ATR period (50 Ma–Present), our Bayesian modelling results show that there is still a common factor with the angiosperms promoting diversification in the post-ATR period, although we also found clade-specific responses (Fig. 4, Supplementary Data 3). The angiosperm effect appears to operate either through a positive correlation with divergence rates (for Diptera and Lepidoptera,) or through a negative correlation with extinction rates (Diptera, Hemiptera, and Hymenoptera). For Coleoptera, we found no significant correlation, but we are very close to inferring a negative correlation with extinction rates during the ATR.

It is also important to highlight that the rise of angiosperms is not the only driver with a significant identified effect on insect evolution. As expected, we also found other factors at play. Notably, a negative diversity-dependent effect of origination rates as a major driver for insects as a whole (Fig. 3, Supplementary Data 1), for pollinator insects only (Fig. 3, Supplementary Data 2), and for the five analysed orders in

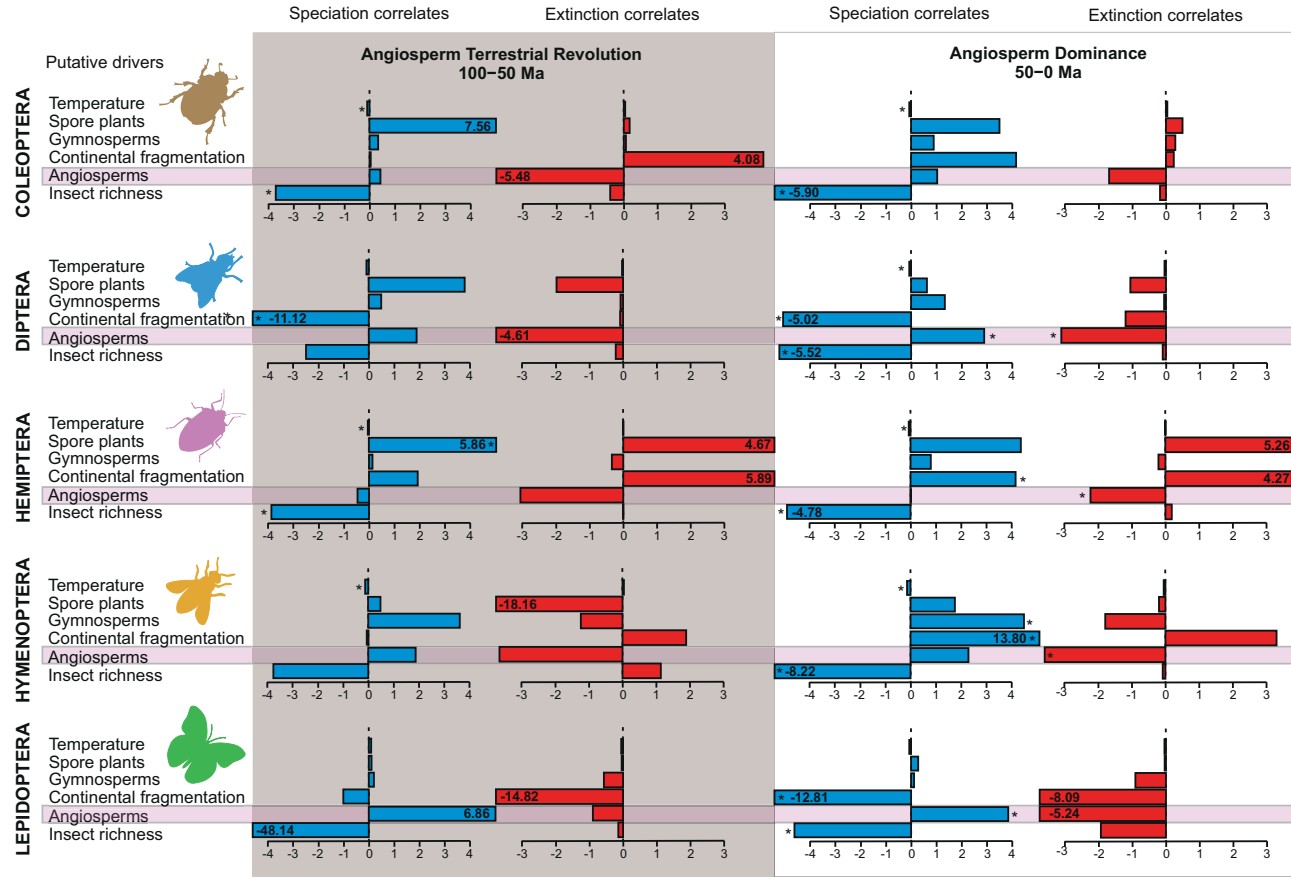

**Fig. 4 | Correlation trends of different analysed drivers for origination (in blue) and extinction (in red) rates on insect diversity divided by five selected orders, for two periods of time: the Angiosperm Terrestrial Revolution timeframe (100–50 Ma[10]), and for the Angiosperm Dominance period (50–0 Ma).** Data used in this representation represent median estimates and the 95% CI is presented in the Supplementary Data 3. Drivers from top to bottom are Global mean temperature (Temperature), Spore plant relative diversity (Spore plants), Gymnosperm relative diversity (Gymnosperms), Continental fragmentation, Angiosperm relative diversity (Angiosperms), and Insect family richness (Insect richness). Asterisks indicate significant correlations recovered with the MBD model (shrinkage weight >0.5 and 95% CI not overlapping with zero). If any of the dates is out of their corresponding scale, it is represented their value inside the box. The x-axes have no unit scale for correlation parameters in the MBD analyses.

the post-ATR period, but also during the ATR for Coleoptera and Hemiptera (Fig. 3, Supplementary Data 3). The third important factor we recovered in the analyses is the effect of past global temperature. It resulted negatively correlated with origination and positively correlated with extinction, such that warmer climates led to lower origination and higher extinction for insects as a whole (Fig. 3, Supplementary Data 1). Global temperature is also found to have a strong negative correlation for the origination rates with pollinator insects in all analyses, indicating lower origination during warmer climates (Fig. 3, Supplementary Data 2). We found that it also influences origination rates through a negative correlation during the ATR (for Coleoptera, Hemiptera, and Hymenoptera) and through a negative correlation with origination rates during the post-ATR period (for all orders except Lepidoptera) (Fig. 4, Supplementary Data 3). Relative spore plant diversity is also found to have a strong positive correlation with both general insect origination and extinction rates, suggesting that higher spore plant diversity spurred insect turnover (Fig. 3, Supplementary Data 1). For pollinator insects, spore plants were only correlated, but strongly, with origination rates (Fig. 3, Supplementary Data 2). Relative gymnosperm diversity is never recovered as a significant driver in the analyses of all insect families, but is found to be an important driver in the analyses of pollinating insect families. Specifically, gymnosperms correlate positively with both origination rates and negatively with extinction rates (Fig. 3, Supplementary Data 2). Relative gymnosperm diversity is also a primer with effect on the origination rates of Hymenoptera during the post-ATR period (Fig. 4, Supplementary Data 3).

## Discussion

Insects are a highly diverse group and are by far the most numerous groups of eukaryotic organisms on Earth[50]. Angiosperms are the most widespread, diverse, and successful groups of extant plants, containing well over 95% of all land plant species alive today[51]. The greatest expansion and diversification of insects began, however, more than 100 million years before angiosperms appear in the fossil record[16,25]. Scientists have long suggested that mutualistic relationships (e.g. pollination) or herbivory between angiosperms and insects may be an important source of insect and plant diversity[52]. This remarkable link between insect origination with angiosperm radiation was most probably the cause that drove the radiation of different groups of herbivores and pollinators such as beetles, bees, and long-proboscid butterflies since the mid-Cretaceous onwards[53–57]. Although shifts in diversification rates are caused by changes in both speciation and extinction rates, it is not trivial to identify the causes of rate shifts in such biological radiation.

On one side, there is the hypothesis that the radiation of insects was not accelerated by the expansion of angiosperms during the Cretaceous (e.g. [16,58]). Hypotheses for the success of insects are rather the evolution of herbivory[59], clade-specific innovations[19], or diversification of parasitic and especially parasitoid insect lineages[21], among others[60]. However, for many of these hypotheses, analyses have shown that the impact of such causes varies greatly between scales and clades, with positive relationships found between the proxies analysed in only selected insect orders, and with low significance or null effects in others. One specific case is the one that refers to parasite and parasitoid insect lineages, because they together are estimated to represent an important amount of the total diversity of extant insects. It has been suggested, however, that the diversification of parasitic and especially parasitoid insect families occurred rapidly during the Late Jurassic–Early Cretaceous and could have been a major driver of the Early Cretaceous peak in family-level insect diversity[21].

On the other side, the close association of flowering plants with pollinating insects has been suggested to have played an important and early role in insect-angiosperm diversification[38,61–64]. This hypothesis has been finally supported in this study (Fig. 3), although there

may be other complementary causes (see also *Results*). Growing evidence from molecular dated phylogenetics, the fossil record of pollinator insects, palaeontological data on plant morphological characters, and modelling of diversification dynamics, supports the hypothesis that angiosperms diversified significantly in the Early Cretaceous, during a period of transient peak in insect diversity[20,21,64] (Fig. 2). This high diversity of insects represents a true burst of origination[21], including insect lineages with highly adapted, pollination modes on gymnosperms[27,36]. This would imply that gymnosperm pollinators were available to angiosperms as they evolved, prior to the first flowering plants. Thus, the co-diversification between insects and angiosperms that we have shown here appears to include, in a wide sense, a pollinator transition of generalist pollen-feeding insects from gymnosperms to angiosperms[32], as already described in different beetle lineages[35,36]. The marginally non-significant negative correlation with extinction rates in Coleoptera with the rise of angiosperms during the ATR seems to support this idea. Similar examples on different insect orders will surely appear soon in the fossil record. Additional alternative hypotheses explaining the radiation of insects are also proposed in the literature (e.g. [12,50]) but are not covered in this analysis, which focuses on the angiosperm-insect co-diversification.

Angiosperm extinction rates decreased after the K/Pg boundary in parallel with increased speciation[6,65], or remained constant throughout the K/Pg event[66], while the opposite is observed for gymnosperm diversity[14,67]. The rise of flowering plants during the Cretaceous, which were twice as productive as gymnosperms[68], was followed by new chemical defence systems, tolerance to climatic stress, and the (genetic) ability of certain angiosperm lineages to repeatedly evolve adaptive traits[69]. These facts, together with climatic changes, the break-up of Pangea, the increase in humid conditions during the Late Cretaceous[70], and the global cooling at the end of the Paleogene, have been linked to the decline in conifer diversity from the Cretaceous (the last 110 Ma) and extending through the Cenozoic[14,71]. The rise of angiosperms led to active displacement by outcompeting conifers[14], and conifers have since experienced high extinction rates[71]. Faced with this situation, gymnosperm pollinators likely had little options but to adapt or go extinct, depending on how specialised they were. The once diverse Cheirolepidiaceae and Bennettitales went extinct around the K/Pg boundary. The extinction of different gymnosperm lineages was followed by the extinction of those insect lineages specifically adapted to these plants, such as highly specialised long-proboscid flies, scorpionflies and lacewings[27,72] (Fig. 1), that could not adapt to new hosts, even for unknown reasons.

Divergent ideas claimed that insect pollination of plants did not guarantee the evolutionary success of the pollinated lineages, because the advantages that early gymnosperm pollinators offered to these plants did not prevent their decline[58]. However, our analyses suggest that this argument should be rejected, because insect pollinators played an important role in the evolution of flowering plants[39]. It would therefore be necessary to analyse the causes of the extinction of numerous gymnosperm lineages since the Cretaceous for other reasons[14,17,71], see above. Our analysis shows that the relative gymnosperm diversity is not recovered as a significant factor to be considered for general insect diversification (Fig. 3, Supplementary Data 1), although it is an important driver for pollinating families (Fig. 3, Supplementary Data 2).

Flowering plants were still low-biomass components of most Cretaceous floras, they did not achieve ecological dominance in a single step[73–75]. It was only after the K/Pg event that the diversification of angiosperms and of insects had explosive effects on each other through their species interactions[25], and only then did angiosperms achieve ecological dominance together with pollinator groups[75,76] and favouring the evolution of multiple new pollinating insect lineages[48,57,77–79] (Figs. 2–4, Supplementary Data 2, 3). In this way, the "modernisation" of many terrestrial ecosystems took place.

We still know little about the origins of entomophily and how it evolved, but theories regarding pollinator-plant coevolution always predict an increased probability of radiation of both plants and the pollinating animals because of the mutualistic nature of the interaction[59]. However, coevolutionary processes should not be considered the only major drivers of diversification in flowering plants and insects[60,80]. Our analyses suggest that global temperature, diversity dependence, and spore plants are additional drivers that should be investigated to explain general insect diversity (Fig. 3, Supplementary Data 1–3). For instance, warmer global temperatures have generally led to lower origination rates and higher extinction rates in all our analyses[81]. Additionally, a negative diversity-dependent effect of origination rates has been recovered for all orders in the post-ATR period, but also during the ATR for Coleoptera and Hemiptera. Spore plants have been also strongly correlated with the origination of pollinator insects. The study of these drivers deserves further analyses in the future besides the effect of angiosperm radiation given the important effect found in our results.

Indeed, insect pollination is not a guarantee of greater success. For example, cycads are insect-pollinated and have never been a diverse lineage[82,83]. Specialisation increases the pollination efficiency, providing a mechanism to explain the increase in speciation rates[84]. However, specialisation also increases the risk of extinction rate under environmental upheavals, as perhaps occurred with highly specialised gymnosperm pollinators. Under fluctuating conditions, plants that are pollinated by specific animals will be more adversely affected than plants that are generally pollinated by multiple species. Most currently known biotically-pollinated plants and their pollinating animals are generalists[85], thus increasing their chances of survival in the face of environmental changes, although insect pollinators that feed on and pollinate a single plant species do exist[58,83]. Careful study of current cases may give us clues about what happened to gymnosperm pollinators and their hosts during the end of the Cretaceous.

## Limitations of the study

We have investigated the deep-time dynamic of insect diversification based on Bayesian inferences of the fossil record. Our approach comes with limitations either related to the dataset or to the methods used. First, it can be difficult to determine whether an insect lineage is a pollinator using only the fossil record, which can hinder our understanding of past diversification for this category of insects. Our study was based on the most recent assessment of insect families recognised as pollinators[28], but we consider our macroevolutionary results to be testable conclusions for future studies. It is likely that some insect lineages, including pollinator lineages, are underrepresented in the fossil record at any time because of their biology, morphology, or different taphonomic biases (e.g., Lepidoptera, Fig. 2b). This effect can hinder our understanding of the past dynamics of insects[84,85]. We cannot overcome this limitation, but new discoveries should also provide additional information on the past diversity of fossil and modern lineages. Although incomplete taxon sampling is pervasive in the fossil record, PyRate can correctly estimate the past dynamics of diversification[41,42]. By running the MBD model over each of the five richest insect orders to (i.e. Coleoptera, Diptera, Hemiptera, Hymenoptera, and Lepidoptera), we show the spectrum of rate heterogeneity and different responses of each group to past environmental upheavals.

Second, as with any process-based model, PyRate makes assumptions about the processes that generate the diversification of a clade. These assumptions are that diversification rates are homogeneous across clades but can vary over time. Therefore, our Bayesian model may violate real evolutionary processes, as there is prior knowledge that insect diversification has varied both across the phylogeny and over time (e.g. [19,47–49]). However, to investigate their evolutionary diversification in relation to our key hypotheses, we considered this rate variation by analysing the five main insect orders separately coupled with a model of rate heterogeneity to account for heterogeneity in the preservation rate across lineages. It remains difficult to estimate diversification rates from the fossil record because it is inherently incomplete and heterogeneous across clades and through time, making any estimates of diversification processes tentative. For instance, the mid-Cretaceous peak in insect diversity followed by an extinction (Fig. 2) has been widely considered as an artefact due to poor preservation[20,21]. Another issue is the poor Maastrichtian fossil record hampering the study of the K/Pg extinction event in insects. However, in our case, it is unlikely that a sudden and strong extinction would alter the estimate of the long-term extinction rates we are interested in. These sampling biases must be accounted for, and PyRate aims to model simultaneously the rates of preservation and their variation across taxa, the times of origin and extinction of each taxon, and then the rates of origination and extinction through time. Using the Bayesian birth-death model with constrained shifts (rates are constant in bins, where bins are the geological epochs), we found that origination and extinction rates do not stand out as outliers in the overall diversification dynamics, despite some large diversity changes in the mid-Cretaceous for example. On the contrary, extinction rates are rather homogeneous and low throughout the Cenozoic and Mesozoic (Supplementary Information Fig. 1). Only the Palaeozoic shows more pronounced rate variations, which could be due to sampling artefacts or the Permian crises[86].

Third, we have considered different taxonomic levels when studying the co-diversification between insects (family level) and plants (genus level). The insect family level was chosen for several reasons: (i) This taxonomic level has been analysed in other studies of fossil diversity[19–21] and appears to correlate well with underlying species diversity[86]; (ii) Families are less susceptible to irregular and biased sampling than species and genera, leading to a better evolutionary signal at this level; (iii) Insect families, especially extant ones, are reasonably well established among researchers, whereas fossil species and genera are more idiosyncratically defined and less likely to correspond to good phylogenetic units; (iv) Insect families individually possess discrete, often highly specialised life histories, and their morphologies directly reflect their trophic guilds, which are informative in diversity studies like here when distinguishing pollinating and non-pollinating families[28]. Analyses at the genus level are in their infancy, and only limited to period of time[86], or specific groups[87], and it would be incredibly complicated to compile all the fossil data with caution at the genus level for all insects throughout the Phanerozoic. By contrast, relative diversity of plants are analysed using genus-level data[6,14], and even at the species level for a given region[88], which provides a more accurate diversity dynamic trends. The use of different taxonomic scales in the MBD analyses in PyRate is likely not ideal but remains the only solution when studying deep-time insect-plant interactions but should not represent an issue when looking at long-term evolutionary trends. Finally, it is important to note that our main result showing the influence of angiosperms may depend on the choice and availability of other environmental variables as alternative drivers. At the global scale, we were only able to focus on six candidates reflecting large-scale environmental changes that could have influenced insect and pollinator diversification. There may be other drivers that deserve attention in the future[12,50].

## Final considerations

The origin of angiosperms, pollinating insects, and their coevolution still remains enigmatic, but significant progress has been made in the last decade with fossil-based and phylogenetic studies. The early diversification of angiosperms and their potential insect pollinators appear to have been largely decoupled[25,40], with pollinator insect lineages predating flowers[25,28,32–34,36]. We also know that the richness of insect families transiently peaked around 125 Ma[20,21], which coincides

with numerous pollinator lineages that were adapted to pollinate gymnosperms at the end of the Early Cretaceous[28,36]. In contrast, most angiosperm families (58–80%) originated between ~100 and 90 Ma, during the warmest phases of the Cretaceous[75]. This is also the exact time when the first angiosperm pollinators are found in the fossil record[28,36]. Despite their age of origin, the rise to ecological dominance of modern angiosperms was geographically heterogeneous and took place over a long period lasting into the Cenozoic, coinciding with the onset of crown diversification in most families[75]. We have found that the angiosperms during the ATR correlated with faster diversification of insects, including pollinators. Analysis of this correlation shows that, in the case of insect pollinators, angiosperms played a dual role that evolved over time, first with a strong extinction mitigating effect in the Cretaceous, which continued but slowed down during the Cenozoic, and then as a driving force for Cretaceous origination, which increased during the Cenozoic, from 50 Ma onwards.

When different pollinator orders are analysed separately, we found clade-specific responses, but there is still a common factor with the angiosperms promoting diversification in the post-ATR period (from the Cenozoic onwards), when angiosperms reached their ecological and geographical dominance. Despite the known biases inherent to each clade, we recovered an important role for the angiosperm radiation in the diversification of all insects, insect pollinators only, and the five most diverse insect orders analysed separately, regardless of their taphonomy (Figs. 3, 4). Reducing the idea of angiosperm-pollinator coevolution to a single period under the analysis of being cause or consequence of each other (e.g. [27]) may be an oversimplification. On the one hand, there was a significant pool of gymnosperm pollinators that may have been available to angiosperms from the beginning[22,28,32,36]. On the other hand, there appears to be a link between the diversification of angiosperms and insects, including pollinators, which is more evident after 50 Ma to Present. Pollination is a very complex system of mutualistic relationships that must be analysed in time, space, and morphology very carefully.

## Methods

### Fossil data and multivariate birth-death analyses

We retrieved the times of origination and times of extinction for 1527 families, including 671 extant and 856 extinct families, which have been estimated from more than 38,000 fossil occurrences at the family level[19]. From this entire insect fossil dataset, we extracted only the pollinator families (including possible pollinator families) based on the most recent work[28], representing 174 families. From the original database, we extracted also the families of the five most diverse insect orders, which correspond to 165 families in Coleoptera, 219 in Hemiptera, 176 in Diptera, 116 in Hymenoptera, and 43 in Lepidoptera. To examine changes in insect family diversity, changes in pollinator family diversity and changes in the family diversity divided by orders through time, we reconstructed the lineages-through-time using PyRate 3[42] using the origination and extinction times of all insect families as input file (-ltt 1 option; R scripts available in the FigShare repository).

A birth-death model (MBD) has been developed and implemented in PyRate to test for a correlation between speciation and extinction rates and changes in environmental variables[43]. We used the MBD model to assess the extent to which biotic and abiotic factors can explain temporal variation in speciation and extinction rates. In the MBD model, speciation and extinction rates can vary through time, but equally across all lineages, through correlations with multiple time-continuous variables, and the strengths and signs (positive or negative) of the correlations are jointly estimated for each variable[43]. The MBD model incorporates temporal fluctuations of environmental variables, so that the speciation and extinction rates can depend on variations in each factor. The correlation parameters can take negative values indicating negative correlation, or positive values for positive correlations. If their value is estimated to be close to zero, no

correlation is estimated. A Markov chain Monte Carlo (MCMC) algorithm jointly estimates the baseline speciation ($\lambda 0$) and extinction ($\mu 0$) rates and all correlation parameters ($G\lambda$ and $G\mu$) using a horseshoe prior to control for over-parameterisation and for the potential effects of multiple testing. The horseshoe prior provides an efficient approach to distinguishing correlation parameters that should be treated as noise (and therefore shrunk around 0) from those that are significantly different from 0 and represent true signal. For all insects, pollinator insects only, and the five insect orders, we ran the MBD model with 50 million MCMC iterations and sampling every 50,000 to approximate the posterior distribution of all parameters ($\lambda 0$, $\mu 0$, ten $G\lambda$, ten $G\mu$, and the shrinkage weights of each correlation parameter, $\omega G$). We summarised the results of the MBD analyses by calculating the posterior median and 95% credible interval of all correlation parameters and the mean of the respective shrinkage weights (over ten replicates), as well as the median and 95% credible interval of the baseline speciation and extinction rates.

The MBD model assumes that diversification rates vary continuously over time with a given variable. However, it is possible for rates to vary positively in one-time interval and negatively in another time interval[89]. In other words, the drivers of diversification may vary over time. We therefore tested whether the impact of angiosperm diversity was similar over time, during the ATR (100–50 Ma, *sensu*[10]) and after the ATR (50–0 Ma). The MBD analyses were thus performed with time constraints to estimate rates within this time interval by setting up the *-maxT 100 minT 50* or *-maxT 50 minT 0* option to represent the ATR and post-ATR periods, respectively. The Python and R scripts are available in the FigShare repository.

### Palaeoenvironmental variables

To identify putative mechanisms of insect diversification, we examined the correlation between a series of past environmental variables and origination/extinction rates over their entire history. We focused on the role of six palaeoenvironmental variables, also called proxies, which have been linked to biodiversity change. These proxies were classified as either abiotic or biotic controls as follows: (i) Biotic controls: Ecological interactions with rapidly expanding clades are increasingly recognised as important macroevolutionary drivers[14,86]. Insects experienced drastic floristic changes throughout their evolutionary history with the origin and rapid radiation of angiosperms at the expense of a decline in diversity of gymnosperms and ferns. The rise and dominance of angiosperms may have contributed to altering the dietary regimes of herbivorous insects, which could in turn have affected insects that depend on herbivores by a cascading effect. We used the relative generic diversity trajectories of angiosperms, gymnosperms, and spore plants (mostly composed of ferns) based on previous estimates of plant diversity that estimated genus-level diversification dynamics of these groups and estimated the times of origination and extinction of all plant genera[6], which allows computing the temporal dynamic of diversity changes and converted into relative generic diversity[14,43]. Note that the relative plant group diversities do not sum to 1. The angiosperm and gymnosperm diversities come from ref. [6], and the spore plant diversity comes from ref. [43]. In the latter study, the authors derived the relative diversities of all plant groups. Their diversities do not add up to 1, probably because they are not all from the same study. Biotic interactions within insects could also have influenced their diversification. For instance, we could draw hypotheses of diversity dependence such that insects could either impact or be impacted by their own diversity. In other words, the change in their diversity can affect their diversification as proposed in diversity-dependent hypotheses. We thus included the palaeodiversity of all insect families to account for diversity dependence within insects as an independent variable. Note that the MBD model does a data scaling (*-r 0* option) with all variables so that the trends in diversity fluctuations are used but not the change in absolute diversity values, which

homogenises the fact that insect diversity and relative plant diversity are for different taxonomic levels. (ii) Abiotic controls: Climate change (variations from warming to cooling periods) is one of the most probable drivers of diversification changes throughout the history of life[14,90]. Major trends in global climate change through time are typically estimated from relative proportions of different oxygen isotopes ($\delta^{18}O$) in samples of benthic foraminifera shells. We merged $\delta^{18}O$ global temperature data from different sources ([91] for the Cenozoic[92]; for the rest of the Phanerozoic) to provide $\delta^{18}O$ data spanning the full time-range over which insect families originated. Second, global continental fragmentation, as approximated by plate tectonic change over time, has also been proposed as a driver of diversity dynamics[44,87]. We retrieved the index of continental fragmentation developed by ref. 44 using palaeogeographic reconstructions for 1-million-year time intervals. This index approaches 1 when all plates are not connected (complete plate fragmentation) and approaches 0 when there is maximum aggregation. All these variables were used as input data for the MBD model. The files for the environmental data are available in the FigShare repository.

## Figures
Figures were created using CorelDRAW Graphics Suite software, version 19.0. (www.coreldraw.com). Fig. 2 was designed based on the results obtained after the data analysis with the software R.

## Reporting summary
Further information on research design is available in the Nature Portfolio Reporting Summary linked to this article.

## Data availability
All data (times of origination and extinction of each insect family) are originally available in the main text or extracted from ref. 19. All data (estimates of origination and extinction rates) generated in this study and source data behind all figures have been deposited in the FigShare digital data repository under accession code [https://doi.org/10.6084/m9.figshare.24076725.v1].

## Code availability
All scripts and codes used in this study have been deposited in the FigShare digital data repository under the accession code [https://doi.org/10.6084/m9.figshare.24076725.v1].

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

## Acknowledgements

We thank Corentin Jouault (Institut de Systématique, Évolution, Biodiversité, France) for his comments on early draft. D.P. thanks the Ministry of Economy and Competitiveness of Spain (project "CRE", Spanish AEI/FEDER, UE CGL2017-84419) and the project 2021SGR-349 ("Sedimentary Geology") from the Secretary of Universities and Research (Government of Catalonia) for financial support. This is contribution no. 4 of the postdoctoral fellowships programme Beatriu de Pinós project 2020 BP 00015, *The flowering plant success—Influence of beetles*, funded to D.P. by the Secretary of Universities and Research (Government of Catalonia) and by the Horizon 2020 programme of research and innovation of the European Union under the Marie-Curie grant agreement no. 801370. F.L.C. is supported by the European Research Council (ERC) under the European Union's Horizon 2020 research and innovation programme (project GAIA, agreement no. 851188).

## Author contributions

D.P. and F.L.B. contributed equally to all aspects of the manuscript.

## Competing interests

The authors declare no competing interests.
