## [Peer Review File · Nature Communications]

The angiosperm radiation played a dual role in the diversification of insects and insect pollinatorsReviewers' Comments:

Reviewer #1:

Remarks to the Author:

In this interesting manuscript, the authors test the impact of multiple historical variables, including diversity of angiosperms, on diversification of insects through the Mesozoic and Cenozoic. They find evidence for a significant impact of past angiosperm diversity on insect speciation and extinction, and for distinct variations of this impact in two time periods (100-50 Ma, 50-0 Ma). While these results are certainly important and worthy of publication, I have some reservations on the manuscript itself as well as the scope of the analyses, which I will attempt to outline below. I hope the authors find these comments helpful.

1. General presentation and discussion: while generally well written, the text would benefit from extra care throughout, with quite a few problematic sentences and typographical errors, and the logic not always clear. Arguably, this is a difficult, very broad topic to write on, and the authors did a great job of taking into account a considerable body of recent (and older) literature on the diversification of both angiosperms (but see below) and insects, making parts of the text read more like a review article. However, one key issue in my opinion is the attempt to tackle a much more ambitious question than that asked by the title and analyses: it is clear throughout the text that the authors also want to discuss the role of insects (and other variables including key traits) on angiosperm diversification. This is particularly clear throughout the discussion, which focusses much more on angiosperm diversification than insect diversification. Clearly, the two are intricately linked (at least for part of their history), as this study contributes to show, but the shift in focus left me rather confused. Hence, given the analyses presented and the primary expertise of both authors, I would recommend to rewrite much of the discussion to focus on insect diversification.

2. Angiosperm questions: despite the remarkable effort to synthesize the complex, and sometimes contradictory literature on the origin and diversification of angiosperms, there are a few important mixups and misrepresentations in various places that call for great caution in revising the text (noted further below under Minor comments).

3. Data and analyses: the entire study is based on three variations of a single analysis using previously published data. I do not think this precludes publication, and the analysis itself is a clever macroevolutionary approach modelling the multivariate impacts of various variables on speciation and extinction rates of the fossil record. Yet I believe incomplete discussion of the merits and limitations of this approach is provided in this manuscript, particularly compared to alternative methods and sources of data:

(a) The entire focus for insects is on family origination and extinction, presumably due to the limitation of the data used. Do we expect the same results as those of a hypothetical (but desirable) analysis that would focus on insect species origination and extinction? And is there a potential inconsistency with using genus-level past diversity curves for plants as explanatory variables? (Note the latter only becomes clear while inspecting the paper by Silvestro et al. 2015, which provided the source data for plants. I think this should be made clear in this new study.)

(b) Curiously, for reasons I do not fully understand, the authors have opted to use relative rather than absolute past (generic) diversity of various plant groups as variables of the model of insect origination and extinction. This requires some justification. Furthermore, because the three relative diversity variables of angiosperms, gymnosperms, and spore plants were concurrently used in the same model, there is a potential risk of interdependence. I do not know whether this is a problem with the model used, but would expect to see this aspect discussed as well.

(c) As the authors are well aware, alternative macroevolutionary methods based on dated phylogenies of extant taxa also exist to address similar questions (including some methods developed by the

second author). These have recently become the topic of a heated debate related to model identifiability, and their ability to accurately estimate extinction rates have also long been questioned. Yet, given their prevalence in the literature, and given the inherent biases and limitations of the fossil record used here, it would seem critical to briefly discuss the merits and limitations of both approaches in this study.

Minor comments

L15: insert "an" before "important" (note there are many other similar minor typos spread across the text, which I won't attempt to list exhaustively here)

L17-18: given that we still have no idea when angiosperms evolved (as in their crown age), I would disagree with this statement

L36-37: many issues with this sentence; first, as ref. 2 showed, support for a pre-Cretaceous crown age of angiosperm has little to do with molecular data themselves; second, crown angiosperms have not been dated as 310-380 Ma in the studies cited (you must be referring to the crown age of seed plants / the stem age of angiosperms here, as in your original preprint)

L39-41: not if we accept a pre-Cretaceous origin of crown angiosperms; ref. 10 definitely did not assert a Cretaceous crown ancestor; suggest updating plausible range to 270-140 Ma as in ref. 2

L47: decline in what?

L52-53: which early-diverging groups are you referring to here? (not clear if insects or plants)

L63: which first plants?; note the first land plants could certainly not be wind-pollinated as they had no pollen (which is a synapomorphy of seed plants)

L65: reword "ancient plants" (too vague)

L72: replace "ginkogaleans" with "ginkgoaleans"?

L76-77: rephrase (not clear)

L82-83: in general, I am not entirely comfortable with the attempt to capture and follow three distinct key periods that previous authors have defined to refer to important aspects of angiosperm diversification (Angiosperm Radiation, Cretaceous Terrestrial Revolution, Angiosperm Terrestrial Revolution), as they might be perceived by some readers as different views on the same process, which I don't think they are. This risk of confusion becomes very apparent in the legend of this (and other figures), which refers to "The period of angiosperm evolution". I find these terms confusing because they could be read as "the period during which crown angiosperms originated" (which we do not know and is definitely older than any of these three intervals) or "the period during which angiosperms have continuously evolved" (which would be the period since their origin to the Present).

L84-85: is the time scale the same for A and B? if so, why is B restricted to the interval from ca. 145-55 Ma?

L98: Condamine et al. (fossil insect dataset) is ref. 19

L112-113: same comment as above (L82-83)

L115-116: only in the fossil record (as the authors know, numerous colleagues believe angiosperms were actually present before the Cretaceous)

L119: Table 1 was not included in the manuscript file provided to me for review

L129: not sure what is meant by "the rise of angiosperms" after the ATR (by the end of the ATR, angiosperms would have long risen by all possible measures)

L140-141: I understand this is the result of the multivariate analysis, but it read strangely to me as relative angiosperm and gymnosperm diversity would be strongly correlated to one another (see my main comment above in point 3c)

L160: replace "superiority" with "advantage"?

L164-167: not sure where this statement comes from; I see that ref. 53 wrote something similar in describing the findings of one earlier study, but the reality is we do not really know and still lack any robust macroevolutionary statistical approaches to measure the impact of singular traits (i.e., those originating only once) on diversification rates

L179-183: hard to follow here (particularly L182, the logic of which is unclear to me)

L183-185: I agree, which is why this whole discussion read very strangely to me (as an attempt to synthesize current knowledge on drivers of diversification in angiosperms, which remains highly incomplete, has now been reviewed more than once in recent literature, and is portrayed rather confusingly in the text here)

L189: there are definitely more key studies to consider here, in addition to ref. 6, in particular the paper by Magallon et al. in *Annals of Botany* 123: 491–503, 2019

L211-212: strange statement

L226-227: I don't think we really know this, given that data still lack on pollination of most species of angiosperms

L235: that peak is not clear to me in Fig. 2 (which shows instead that insect family richness has never been so high than in the last 40 Ma)

L245: ref. 10 drafted the angiosperm spindle as hypothetical and largely unknown (see legend of their Fig. 1), which I think should be noted here as well if reproduced; on the other hand, ref. 10 did certainly not depict detailed spindles of the various gymnosperm lineages, so the legend and citation here must be corrected

L247-248: I am not familiar with ref. 99, but find it very odd and confusing to still recognise here the possibility of Gnetales being closely related to angiosperms (as denoted by the hyphenated branch), an old idea now dismissed by most phylogenetic analyses published over the last two decades

L248: I have reservations on the use of TimeTree (which averages over all studies in their database, regardless of quality and assumptions), but I suppose accurate divergence times (and their uncertainty) are not critical to either this figure or this study

L260: strange sentence, needs rewording

L270-273: I don't disagree, but this is a rather underwhelming closing statement in that various authors have already suggested the same and it remains somewhat disconnected from this study (perhaps some rewriting needed)

L279: wrong Condamine et al. paper cited here? Shouldn't this be ref. 19 instead?

L510: first author should be Lagomarsino

Hervé Sauquet

Reviewer #2:

Remarks to the Author:

This manuscript addresses drivers of insect diversification during the Cretaceous and Cenozoic at the family level using fossil data. It uses a Bayesian approach applied to fossil origination and extinction data to develop and assess birth-death models and parameterize correlations between those births and deaths and other continuously varying ecological parameters. These include the richness of different taxa, including insect families and the richness of different plant groups, plus sea surface temperature estimates and the fragmentation of continents. Models are developed for two 50Myr time periods to assess to what extent the associations have varied over time, and these time periods are chosen to match periods described as the Angiosperm Terrestrial revolution (angiosperms replacing other plants, notably Gymnosperms), and the later period of Angiosperm dominance. The authors report that insect origination and extinction are explained by a number of variables, but not gymnosperm diversity. Origination rates are enhanced by spore plant diversity, angiosperm diversity (but only in the later time period) and reduced by insect diversity (implying diversity-dependence). Extinction rates are also enhanced by spore plant diversity, continental fragmentation in the first time period, and reduced by angiosperm diversity. The authors conclude a changing role for angiosperms affecting insect diversity in association with several other variables.

The questions addressed by this work are of wide appeal: angiosperms and insects collectively comprise about three quarters of extant macroscopic species described on Earth, and because insects and plants interact, there has long been speculation on how they could have affected each other's macroevolution. Fossil data provide some advantages over purely phylogenetic data based on extant taxa as they are probably better sources for estimating extinction rates. The analytical methods used are current and appropriate for the data. The results of the paper are interesting as they are the first to my knowledge which explicitly test several potential environmental drivers of insect diversity at a global scale, and test how those change over time. The findings should contribute to the developing story of how the global taxonomic composition and richness changes of the last 100Myr came about.

Whilst I think there is the backbone of an interesting paper here, I also feel that there are some significant issues for the authors to consider to make it more interesting and robust. These relate to the way the story is developed in the introduction and discussion, the scope of the analyses performed and how those relate to the story, and the sensitivity of the results to potential biases and variations in the data. There are also some textual/content issues. I list these below. I hope they are useful to the authors in developing this work further.

1. Table 1 is mentioned several times in the main text but seemed to be absent from the manuscript. Since this seems to contain important results, it's difficult to make a full decision on the manuscript.
2. The introduction and discussion seem to mismatch the results and methods quite significantly. This makes a confusing story for the reader. Specifically, large parts of the introduction and discussion are spent developing a story about plant pollinator interactions. For example, Figure 1 is all about pollinators and their interactions with angiosperms or gymnosperms. This is fair enough and interesting. However, it is then a bit surprising to find that the data and results cover all insects, not just pollinators. If I were testing an hypothesis about plant pollinator interactions, I would design my data around those groups. The discussion is extensively about pollinator interactions too, but of course a vast number of the species included in the families covered by the data are not pollinators at all. Plenty of the species and families do not directly interact with plants: granted most probably do

indirectly, which is relevant to a general insect-angiosperm story, but this is hardly mentioned. In addition there are a great number of ways insects interact with plants that do not include pollination, herbivory being the most obvious and probably important. There is almost nothing on herbivory here! Maybe the focus needs also to consider larval as well as adult interactions. This seems like a rather large oversight, or at least it is confusing for the reader. I think that the authors should decide if they want to test ideas about pollination or a much more general set of ideas about how plants affect insects, but whatever they choose the story should match the data and analyses.

3. Similar to the above there is a focus to some extent in the introduction and especially in the discussion on angiosperm diversification, but the results and analyses focus exclusively on how angiosperms affect insect macroevolution. It's almost as if the results were written by someone with interests in insects but discussion by someone with interests in angiosperms. Now, given that the term co-diversification or co-evolution is used liberally, I was expecting analyses on how insects affect angiosperms as well as vis-versa. The data imply that this could be done, so I was wondering why not. The results are better suited to a narrative exclusively focussed on insects and not focusing on angiosperm macroevolution.

4. Insect fossils come from a variety of sources but two main sources are impression fossils such as from lake sediments, and specimens preserved in amber. These are two very different types of preservation, with the latter being much more sparse and episodic but giving much better sampling (and higher implied richness). The PyRate method models preservation changes but it would seem a bit risky to base all conclusions on a mixed dataset that combines these two sources of evidence because the preservation characteristics are so different. It would be sensible I think to conduct a sensitively analysis by doing additional analyses on a single (e.g. non-amber) subset or if possible both subsets (amber and non-amber) to see how robust the results are to this variation in data sources: this has been done for example in ref 20, and it should be straightforward to extract the data from PBDB. Similarly, some exploration of taxonomic subsets would be very sensible given that different trophic groups would be expected to have more direct or indirect associations with angiosperms.

5. The discussion hardly discusses your actual findings at all. Why should higher angiosperm diversity reduce extinction rates in insects? Has any literature proposed this before? There is a vast literature on density-dependent diversification in insects but not much has been cited: for example see the papers discussed in the reviews by Mayhew in 2007 (Biol. Rev.) and 2018 (Ent Exp et Appl). Are you surprised that it seems to work via reductions in origination rather than rises in extinction? Would you expect continental fragmentation to have the observed effect on extinction and why? Given your hypotheses are you surprised by the non-effect of gymnosperm diversity on extinction? Could a subset of older families show different results as in figure 1? Could you test that? The temperature effects conform to some previous results on invertebrates but not all. The contrary papers are not mentioned. Why might they exist?

6. The whole issue of working at the family level is not discussed anywhere but I think the reasons and caveats need to be mentioned somewhere, at least in the discussion. For example, extinction of a family implies very different species level rates for some families than others. Very species rich families will be almost impossible to send extinct, whilst species poor ones might go extinct easily. Is it likely that species richness across families has shown systematic trends over time? Are family level data in plants and insect comparable and the most relevant data in which to make these comparisons? In reality, we are simply using this level in order to reduce gaps in the record. Why would one expect family level diversity in plants to have any effect on family level diversity in insects: what assumptions are built into that?

7. Another caveat is that the PyRate models assume that rates are the same at any moment in time in all lineages. Of course we know that taxon-specific rates exist in insects. What are the consequences of this breakage of assumptions for accurately estimating rates? I know the methods are what we have, but the reader needs to be alerted to caveats.

8. If you keep the existing text and figures, I have minor suggested edits:

- i) Line 15 drivers
- ii) Line 47 delete and
- iii) Line 77 using angiosperm resources extensively until the

- iv) Line 131: dual influence?
- v) Line 134: Nevertheless, the rise of
- vi) Line 136: correlated with reduced origination rates
- vii) Line 145: focussed on the
- viii) Figure 3 legend and axis labels: No units or title on the x-axis scale: from the main text one would naively assume this to be a correlation coefficient but it cannot be. Why not have the confidence intervals on the bars?
- ix) Line 172: cell sizes; herbaceousness,
- x) Line 185: approach that
- xi) I felt Figure 4 was not needed; it's not results and this is not a review paper.
- xii) Line 260: found that flowering plant diversity correlates with a faster
- xiii) Line 266-7. Other hand, a link seems to exist between
- xiv) Line 335: need to say what taxonomic level these data exist at.
- xv) Line 367. I didn't have access to the data, but they certainly weren't in the main text: supplementary materials?

Reviewer #3:

Remarks to the Author:

This manuscript investigates the potential effects of flowering plants on insect diversification. The link between angiosperms and insects has been hypothesized and studied for quite some time, prompted by the many ecological relationships through pollination and other interactions. Given the high diversity of insects, it is natural that this is a topic of widespread interest.

The methods calculate origination, extinction, and diversity from the fossil record, but the techniques do not appear to account for sampling biases or methodological shortcomings of range-through diversity methods. Most notable is the "pull of the recent," which is visible in the family accumulation plot in figure 2. When information from the extant insect fauna is combined with fossil data, this has the effect of inflating diversity and reducing origination and extinction in more recent time periods. (Technically, it is older time periods that have artificially-low diversity because they do not benefit from the range extensions caused by including extant data.)

Because the analysis didn't account for specific challenges of the fossil record, many of the conclusions about the role of angiosperms could instead be spurious correlations between the angiosperm dominance time series and the pull of the recent. Likewise, reduced origination during times of high diversity (lines 135-137) could be diversity dependence, but also could reflect the pull of the recent bias. It is possible that there are true effects, but I don't think the argument is convincing now because the methods are not appropriate for dealing with fossil data, and do not attempt to discuss or address widely-known biases.

A lot of the discussion section seemed to be on topics that are tangential to the results, rather than making the case that the results are true biological patterns, or explaining why the drivers had an effect on insect diversification. For example, why might spore plants also have influenced insect diversification? A lot of the discussion reads like background information about angiosperm evolution, which is interesting, but I didn't think it was strongly connected to your specific goal in this manuscript.

In general, I think it is extremely difficult to work with origination and extinction in the insect fossil record (even harder than trying to reconstruct diversity, which itself is very challenging). Shifts in the dominant preservation mode (amber vs. compression fossils) from one time period to the next cause apparent origination/extinction spikes because the two modes tend to record different types of families. There are huge variations in the number of insects recorded in different time periods, including very few large localities between about 100 Ma and 50 Ma. Because of that, a lot of

extinctions that actually occurred later in the Cretaceous will all cluster earlier. Originations will also be artificially clustered at some of the "super-Lagerstätte" (the extraordinarily well-sampled Baltic amber is one example; likely a number of those families evolved earlier but have not yet been discovered because of the limited Late Cretaceous and Paleocene record). It may be possible to extract some real signal, but I'm fairly skeptical, and it would require careful analysis that deals with the complexity of the insect fossil record.

It was also difficult to evaluate the results because I couldn't find table 1 and there didn't appear to be a dataset containing the potential drivers (spore plant, gymnosperm, angiosperm dominance, temperature, continental fragmentation). Perhaps I missed those on the website, but it would have been very helpful to see a graph of angiosperm dominance through time so that I could visually compare it to insect diversification.

Sincerely,
Matthew Clapham

RESPONSE TO REVIEWERS' COMMENTS

Reviewer #1 (Remarks to the Author):

In this interesting manuscript, the authors test the impact of multiple historical variables, including diversity of angiosperms, on diversification of insects through the Mesozoic and Cenozoic. They find evidence for a significant impact of past angiosperm diversity on insect speciation and extinction, and for distinct variations of this impact in two time periods (100-50 Ma, 50-0 Ma). While these results are certainly important and worthy of publication, I have some reservations on the manuscript itself as well as the scope of the analyses, which I will attempt to outline below. I hope the authors find these comments helpful.

Thank you for your thorough review and overall positive insights. We appreciate all the comments and have addressed all of them below.

1. General presentation and discussion: while generally well written, the text would benefit from extra care throughout, with quite a few problematic sentences and typographical errors, and the logic not always clear. Arguably, this is a difficult, very broad topic to write on, and the authors did a great job of taking into account a considerable body of recent (and older) literature on the diversification of both angiosperms (but see below) and insects, making parts of the text read more like a review article. However, one key issue in my opinion is the attempt to tackle a much more ambitious question than that asked by the title and analyses: it is clear throughout the text that the authors also want to discuss the role of insects (and other variables including key traits) on angiosperm diversification. This is particularly clear throughout the discussion, which focusses much more on angiosperm diversification than insect diversification. Clearly, the two are intricately linked (at least for part of their history), as this study contributes to show, but the shift in focus left me rather confused. Hence, given the analyses presented and the primary expertise of both authors, I would recommend to rewrite much of the discussion to focus on insect diversification.

We understand this point. Perhaps the issue comes from the fact that our first attempt to submit a manuscript to *Nature Communications* included a review of both plant and insect life history and evolution. However, following the indications from the editorial office, we had to delete the review section and focus on the analyses and their results. During the revision, we have tried to create a fresh thread in the manuscript.

2. Angiosperm questions: despite the remarkable effort to synthesize the complex, and sometimes contradictory literature on the origin and diversification of angiosperms, there are a few important mixups and misrepresentations in various places that call for great caution in revising the text (noted further below under Minor comments).

Thank you for this note. We have considered the cited comment in the revised manuscript if more detailed data is provided below.

3. Data and analyses: the entire study is based on three variations of a single analysis using previously published data. I do not think this precludes publication, and the analysis itself is a clever macroevolutionary approach modelling the multivariate impacts of various variables on speciation and extinction rates of the fossil record. Yet I believe incomplete discussion of the merits and limitations of this approach is provided in this manuscript, particularly compared to alternative methods and sources of data:

(a) The entire focus for insects is on family origination and extinction, presumably due to the limitation of the data used. Do we expect the same results as those of a hypothetical (but desirable) analysis that would focus on insect species origination and extinction? And is there a potential inconsistency with using genus-level past diversity curves for plants as explanatory variables? (Note the latter only

becomes clear while inspecting the paper by Silvestro et al. 2015, which provided the source data for plants. I think this should be made clear in this new study.)

An analysis focusing on insect species (or even genera) from the fossil record would take several years to complete. We have considered using families as a representation of origination and extinction rates in insect evolution. Of course, we agree that finest analyses (at lower taxonomic levels) would provide a more robust signal, but the results are highly unlikely to reach the reviewer's wish at this point. Genus-level fossil-based analyses are only starting to be published, but for a limited timeframe, as it is incredibly complicated to compile all the data with caution (Jouault et al. 2022 – *Nat. Comm.*).

(b) Curiously, for reasons I do not fully understand, the authors have opted to use relative rather than absolute past (generic) diversity of various plant groups as variables of the model of insect origination and extinction. This requires some justification. Furthermore, because the three relative diversity variables of angiosperms, gymnosperms, and spore plants were concurrently used in the same model, there is a potential risk of interdependence. I do not know whether this is a problem with the model used, but would expect to see this aspect discussed as well.

Thank you for this point. It would be ideal to rely on the 'absolute past diversity' for each plant group, but it's currently impossible to get descent estimates for each of these clades. To our knowledge, the most recent and updated estimates of past diversity for these three groups is Silvestro *et al.* (2015 – *New Phytol.*). This study still provides rough estimates of the past diversity of plants, and as such we prefer to rely on relative diversity rather than absolute diversity. That being said, note that the curves of both diversity estimates will be very similar, but one is standardized from 0 to 1 (relative), while the other represents number of genera (absolute). We have now better explained the rationale behind the use of relative diversity in our analyses in the *Methods* section.

(c) As the authors are well aware, alternative macroevolutionary methods based on dated phylogenies of extant taxa also exist to address similar questions (including some methods developed by the second author). These have recently become the topic of a heated debate related to model identifiability, and their ability to accurately estimate extinction rates have also long been questioned. Yet, given their prevalence in the literature, and given the inherent biases and limitations of the fossil record used here, it would seem critical to briefly discuss the merits and limitations of both approaches in this study.

Thank you for this other good point. It's not a simple one to address here but we agree with the referee that alternative modelling approaches do exist to estimate rates of diversification. The fossil record is one way, but molecular dated phylogenies is another one. It is true that phylogenetic birth-death models are now widespread to test macroevolutionary hypotheses and that there is major debate in the reliability of rate estimates from them that stem from Kubo & Iwasa (1995 – *Evolution*) and reanimated by Louca & Pennell (2020 – *Nature*), including many studies in between. We would have loved to estimate diversification rates from the latest family-level insect dated phylogeny (certainly Rainford *et al.* 2014 – *PloS One*) if a similar phylogenetic birth-death model would have existed to test multiple variables simultaneously. However, we agree that it is critical to discuss the merits and limitations of both approaches in this study, which we have done in the revised version.

Minor comments

L15: insert "an" before "important" (note there are many other similar minor typos spread across the text, which I won't attempt to list exhaustively here)

Added. We have paid attention to remove typos and grammatical errors from the text during the revision.

L17-18: given that we still have no idea when angiosperms evolved (as in their crown age), I would disagree with this statement

We agree that angiosperm age is difficult to circumscribe. However, the corresponding sentence is based on the article by Benton *et al.* (2022 – *New Phytol.*), where the reviewer signs as an author, and where it is established that the terrestrial revolution produced by angiosperms begins 100 million years

ago (called the *Angiosperm Terrestrial Revolution*). In the text we cite the controversy regarding the paleontological and molecular data for the evolution of angiosperms, but this is just the abstract. Nonetheless, we have substantially rephrased the abstract.

L36-37: many issues with this sentence; first, as ref. 2 showed, support for a pre-Cretaceous crown age of angiosperm has little to do with molecular data themselves; second, crown angiosperms have not been dated as 310-380 Ma in the studies cited (you must be referring to the crown age of seed plants / the stem age of angiosperms here, as in your original preprint)

We agree. "This age is almost entirely conditioned by its own prior distribution" (Sauquet *et al.* 2022 – *J. Exp. Bot.*, ref. 2 in the text). Because of that we included ref. 2 first. But there are different results that we must cite (even that conditioned), and in all of them they use molecular support and get similar pre-Cretaceous origin for the **STEM** angiosperms (these last words changed accordingly, thanks). We showed again the contrast with fossils including the ref. 8, showing first evidence of tricolpate pollen 121 Ma only, followed by ref- 9–11.

L39-41: not if we accept a pre-Cretaceous origin of crown angiosperms; ref. 10 definitely did not assert a Cretaceous crown ancestor; suggest updating plausible range to 270-140 Ma as in ref. 2

The sentence "The age of crown Angiospermae, variously dated from 250–140 Ma, remains controversial and a matter of intense research from both palaeobotanical and fossil-calibrated molecular dating approaches" has been extracted from ref. 10. (Benton *et al.*, 2022). As we indicated in our manuscript. However, we slightly changed our word to be clearer in the new text.

L47: decline in what?

Sorry for the confusion. It was a decline in conifers. It was accidentally missing after the edition following the editor instructions. Now solved.

L52-53: which early-diverging groups are you referring to here? (not clear if insects or plants)

It is referring to the insect peak cited in the previous sentence: "*Insect family-richness peaked during the Early Cretaceous around 125 Ma, when angiosperms were still rare [20–21]. This peak occurred...*" We tried to be more specific now to avoid confusion.

L63: which first plants?; note the first land plants could certainly not be wind-pollinated as they had no pollen (which is a synapomorphy of seed plants)

Correct, thank you for the nuance. We included "first seed plants".

L65: reword "ancient plants" (too vague)

We changed it by "Cretaceous plants".

L72: replace "ginkogaleans" with "ginkgoaleans"?

Thank you. Changed.

L76-77: rephrase (not clear)

Done: "*But insects not only pollinated gymnosperms until, at least, the Late Cretaceous [32, 35–36], they continue pollinating flowering plants since their emergence [32, 35, 38–39].*"

L82-83: in general, I am not entirely comfortable with the attempt to capture and follow three distinct key periods that previous authors have defined to refer to important aspects of angiosperm diversification (Angiosperm Radiation, Cretaceous Terrestrial Revolution, Angiosperm Terrestrial Revolution), as they might be perceived by some readers as different views on the same process, which I don't think they are. This risk of confusion becomes very apparent in the legend of this (and other figures), which refers to "The period of angiosperm evolution". I find these terms confusing because they could be read as "the period during which crown angiosperms originated" (which we do not know

and is definitely older than any of these three intervals) or “the period during which angiosperms have continuously evolved” (which would be the period since their origin to the Present).

We agree with the reviewer about the possible confusion and have therefore changed “*the period of angiosperm evolution*” to “*the period of increase in terrestrial biodiversity*” for clarity. However, we have kept the three periods in the figure, as they show different concepts depending on the reference.

L84-85: is the time scale the same for A and B? if so, why is B restricted to the interval from ca. 145-55 Ma?

Yes, it is the same scale. It is like that because were original data extracted from Peris et al. (2017) ref. 35. We clarified that: “*Data extracted from [35].*”

L98: Condamine et al. (fossil insect dataset) is ref. 19

Corrected. Thank you.

L112-113: same comment as above (L82-83)

Also changed following the same answer.

L115-116: only in the fossil record (as the authors know, numerous colleagues believe angiosperms were actually present before the Cretaceous)

Correct. We included this comment: “*for which angiosperms were absent in the fossil record until the Early Cretaceous*”.

L119: Table 1 was not included in the manuscript file provided to me for review

We are sorry for this issue. Because we made different attempts to submit the manuscript to the journal perhaps the table was misplaced at some point of the process. We have now fixed this in the revised version.

L129: not sure what is meant by “the rise of angiosperms” after the ATR (by the end of the ATR, angiosperms would have long risen by all possible measures).

We meant the time of the ATR itself. We changed it to be clearer.

L140-141: I understand this is the result of the multivariate analysis, but it read strangely to me as relative angiosperm and gymnosperm diversity would be strongly correlated to one another (see my main comment above in point 3c)

Sorry for the confusion. We have indeed performed multivariate analyses with 6 variables, of which the relative diversities of angiosperms, gymnosperms, and spore plants were included as explanatory variables in the model. Our analyses do not find evidence for correlation between the insect diversification and gymnosperm relative diversity. We could not assess the correlation between angiosperms and gymnosperms in this study with the data in hand.

L160: replace “superiority” with “advantage”?

Done.

L164-167: not sure where this statement comes from; I see that ref. 53 wrote something similar in describing the findings of one earlier study, but the reality is we do not really know and still lack any robust macroevolutionary statistical approaches to measure the impact of singular traits (i.e., those originating only once) on diversification rates

Exactly, and instead of these singular traits, “*intrinsic key innovations, extrinsic factors such as geography and environment, or trait–environment combinations are important drivers of diversification rates*” (refs. 59 and 60).

L179-183: hard to follow here (particularly L182, the logic of which is unclear to me)

The sentence has been rephrased in the reviewed version.

L183-185: I agree, which is why this whole discussion read very strangely to me (as an attempt to synthesize current knowledge on drivers of diversification in angiosperms, which remains highly incomplete, has now been reviewed more than once in recent literature, and is portrayed rather confusingly in the text here).

We hope the discussion was clearer after including the suggested modifications along the text.

L189: there are definitely more key studies to consider here, in addition to ref. 6, in particular the paper by Magallon et al. in *Annals of Botany* 123: 491–503, 2019

We agree. The literature on this topic is dense. We have included the proposed reference.

L211-212: strange statement

We referred to the fact that what we are analyzing is what originated the modern ecosystems on Earth.

L226-227: I don't think we really know this, given that data still lack on pollination of most species of angiosperms

We changed it including "known".

L235: that peak is not clear to me in Fig. 2 (which shows instead that insect family richness has never been so high than in the last 40 Ma)

The appreciation that insect family richness is greater in last 40 Ma is correct, but there is a peak undoubtedly 125 Ma to which we referred along the text in different sections.

L245: ref. 10 drafted the angiosperm spindle as hypothetical and largely unknown (see legend of their Fig. 1), which I think should be noted here as well if reproduced; on the other hand, ref. 10 did certainly not depict detailed spindles of the various gymnosperm lineages, so the legend and citation here must be corrected

Information updated.

L247-248: I am not familiar with ref. 99, but find it very odd and confusing to still recognise here the possibility of Gnetales being closely related to angiosperms (as denoted by the hyphenated branch), an old idea now dismissed by most phylogenetic analyses published over the last two decades.

The figure has been deleted in the new version, following recommendations from another reviewer.

L248: I have reservations on the use of TimeTree (which averages over all studies in their database, regardless of quality and assumptions), but I suppose accurate divergence times (and their uncertainty) are not critical to either this figure or this study.

We personally agree with the reviewer in all this content.

L260: strange sentence, needs rewording

We have reworded the sentence, thank you for the suggestion.

L270-273: I don't disagree, but this is a rather underwhelming closing statement in that various authors have already suggested the same and it remains somewhat disconnected from this study (perhaps some rewriting needed)

We deleted this last sentence.

L279: wrong Condamine et al. paper cited here? Shouldn't this be ref. 19 instead?

Yes, thank you.

L510: first author should be Lagomarsino

Changed.

Hervé Sauquet

Reviewer #2 (Remarks to the Author):

This manuscript addresses drivers of insect diversification during the Cretaceous and Cenozoic at the family level using fossil data. It uses a Bayesian approach applied to fossil origination and extinction data to develop and assess birth-death models and parameterize correlations between those births and deaths and other continuously varying ecological parameters. These include the richness of different taxa, including insect families and the richness of different plant groups, plus sea surface temperature estimates and the fragmentation of continents. Models are developed for two 50Myr time periods to assess to what extent the associations have varied over time, and these time periods are chosen to match periods described as the Angiosperm Terrestrial revolution (angiosperms replacing other plants, notably Gymnosperms), and the later period of Angiosperm dominance. The authors report that insect origination and extinction are explained by a number of variables, but not gymnosperm diversity. Origination rates are enhanced by spore plant diversity, angiosperm diversity (but only in the later time period) and reduced by insect diversity (implying diversity-dependence). Extinction rates are also enhanced by spore plant diversity, continental fragmentation in the first time period, and reduced by angiosperm diversity. The authors conclude a changing role for angiosperms affecting insect diversity in association with several other variables.

The questions addressed by this work are of wide appeal: angiosperms and insects collectively comprise about three quarters of extant macroscopic species described on Earth, and because insects and plants interact, there has long been speculation on how they could have affected each other's macroevolution. Fossil data provide some advantages over purely phylogenetic data based on extant taxa as they are probably better sources for estimating extinction rates. The analytical methods used are current and appropriate for the data. The results of the paper are interesting as they are the first to my knowledge which explicitly test several potential environmental drivers of insect diversity at a global scale, and test how those change over time. The findings should contribute to the developing story of how the global taxonomic composition and richness changes of the last 100Myr came about. Thank you for your review and the positive assessment. Everything is clear up to this point.

Whilst I think there is the backbone of an interesting paper here, I also feel that there are some significant issues for the authors to consider to make it more interesting and robust. These relate to the way the story is developed in the introduction and discussion, the scope of the analyses performed and how those relate to the story, and the sensitivity of the results to potential biases and variations in the data. There are also some textual/content issues. I list these below. I hope they are useful to the authors in developing this work further.

1. Table 1 is mentioned several times in the main text but seemed to be absent from the manuscript. Since this seems to contain important results, it's difficult to make a full decision on the manuscript. This is correct, we are sorry for this error. It was misplaced at some point of the submitted process, which we did different times for some reasons.

2. The introduction and discussion seem to mismatch the results and methods quite significantly. This makes a confusing story for the reader. Specifically, large parts of the introduction and discussion are spent developing a story about plant pollinator interactions. For example, Figure 1 is all about pollinators and their interactions with angiosperms or gymnosperms. This is fair enough and interesting. However, it is then a bit surprising to find that the data and results cover all insects, not just pollinators. If I were testing an hypothesis about plant pollinator interactions, I would design my data around those groups. The discussion is extensively about pollinator interactions too, but of course a vast number of the species included in the families covered by the data are not pollinators at all. Plenty of the species and families do not directly interact with plants: granted most probably do indirectly, which is relevant to a general insect-angiosperm story, but this is hardly mentioned. In addition there are a great number of ways insects interact with plants that do not include pollination,

herbivory being the most obvious and probably important. There is almost nothing on herbivory here! Maybe the focus needs also to consider larval as well as adult interactions. This seems like a rather large oversight, or at least it is confusing for the reader. I think that the authors should decide if they want to test ideas about pollination or a much more general set of ideas about how plants affect insects, but whatever they choose the story should match the data and analyses.

We agree with these comments, it's fair. It was our original intention to focus on insect pollinator but we struggled to test our hypotheses using only pollinator groups because of the lack of definition for insect pollinator groups focused in the fossil record. This was the reason why we used all insect data available as a proxy for insect pollination. However, in the meantime, a publication arose reviewing the pollinator habits from insect groups (Peña-Kairath *et al.* 2023 – *Trends Ecol. Evol.*), co-authored by the first author of this current study. During the revision, we have used data from this study to categorize insect families as pollinators and we have then performed new diversification analyses to present a more focused analysis in the new version of the manuscript to fit better with our introduction, discussion, and figures about pollinators. Nonetheless, we think it's important to keep the results for insect families as a whole because, as you mentioned, this is relevant to the study of insect-plant interactions and for comparison with the pollinators themselves.

3. Similar to the above there is a focus to some extent in the introduction and especially in the discussion on angiosperm diversification, but the results and analyses focus exclusively on how angiosperms affect insect macroevolution. It's almost as if the results were written by someone with interests in insects but discussion by someone with interests in angiosperms. Now, given that the term co-diversification or co-evolution is used liberally, I was expecting analyses on how insects affect angiosperms as well as vis-versa. The data imply that this could be done, so I was wondering why not. The results are better suited to a narrative exclusively focussed on insects and not focusing on angiosperm macroevolution.

We do agree that our results are better suited to a narrative exclusively focused on insect macroevolution and not on angiosperm macroevolution. The text was structured differently. We hope this version will be liked by the reviewer.

We do not use co-diversification and co-evolution liberally, but on the contrary. We followed strictly the original description of the idea of co-evolution, but addressing this issue from co-diversification, which it is the unique results that we can offer. We try to use the correct term at all times, co-evolution for the concept, co-diversification for our results.

4. Insect fossils come from a variety of sources but two main sources are impression fossils such as from lake sediments, and specimens preserved in amber. These are two very different types of preservation, with the latter being much more sparse and episodic but giving much better sampling (and higher implied richness). The PyRate method models preservation changes but it would seem a bit risky to base all conclusions on a mixed dataset that combines these two sources of evidence because the preservation characteristics are so different. It would be sensible I think to conduct a sensitively analysis by doing additional analyses on a single (e.g. non-amber) subset or if possible both subsets (amber and non-amber) to see how robust the results are to this variation in data sources: this has been done for example in ref 20, and it should be straightforward to extract the data from PBDB. Similarly, some exploration of taxonomic subsets would be very sensible given that different trophic groups would be expected to have more direct or indirect associations with angiosperms.

In order to make the results more in line with our original idea, as the reviewer suggested earlier, we have selected the families that are pollinators (or were pollinators at some point in their evolution). However, although the proposed complementary analyses suggested by the reviewer will certainly be very fruitful, we feel that it goes beyond the aim of our work. We do not intend to analyse taphonomic issues, which have been addressed in ref. 20 and others, but the relationship between plants and pollinator insects. However, even if we were to follow the reviewer's suggestions, it would mean that we would have to completely rebuild the database used for the study in order to be able to include the type of fossil from which each record was obtained, since some are in amber, some in compression,

and some in both. This work and the combinations it would involve would be the subject of a new study by itself, as was done ref. 20 cited in the text.

5. The discussion hardly discusses your actual findings at all. Why should higher angiosperm diversity reduce extinction rates in insects? Has any literature proposed this before? There is a vast literature on density-dependent diversification in insects but not much has been cited: for example see the papers discussed in the reviews by Mayhew in 2007 (Biol. Rev.) and 2018 (Ent Exp et Appl). Are you surprised that it seems to work via reductions in origination rather than rises in extinction? Would you expect continental fragmentation to have the observed effect on extinction and why? Given your hypotheses are you surprised by the non-effect of gymnosperm diversity on extinction? Could a subset of older families show different results as in figure 1? Could you test that? The temperature effects conform to some previous results on invertebrates but not all. The contrary papers are not mentioned. Why might they exist?

We think that the reviewer has very high expectations of the results that we can discuss in this paper. The discussion we offer was intended to be focused on the relationship of pollinators with plants. We do not analyze the rest of the factors, but we simply show that the one we are interested in have an effect, but we also want to mention that there are other factors that should be considered in the future, as we commented in the conclusion. Because the journals such as NC are subject to very strict content rules (such as the number of words, references, etc.), it is difficult to satisfy all the indications at the same time when some of them are contrary to each other. The reference Mayhew in 2007 (Biol. Rev.) has been included in the list and properly cited along the text.

6. The whole issue of working at the family level is not discussed anywhere but I think the reasons and caveats need to be mentioned somewhere, at least in the discussion. For example, extinction of a family implies very different species level rates for some families than others. Very species rich families will be almost impossible to send extinct, whilst species poor ones might go extinct easily. Is it likely that species richness across families has shown systematic trends over time? Are family level data in plants and insect comparable and the most relevant data in which to make these comparisons? In reality, we are simply using this level in order to reduce gaps in the record. Why would one expect family level diversity in plants to have any effect on family level diversity in insects: what assumptions are built into that?

The first reason to work with families and not with genera or species is practical. It results highly difficult to manage data from around 1500 insect species. But if we referred to the about 1 million named insect species (Stork, 2018, Annual Review of Entomology) it will be impossible to manage.

In addition, errors derived from synonymy or name changes between species and genera, and even more so when we refer to the fossil record, are reduced if we deal with families than in lower taxonomic levels.

After these two considerations, it will result very interesting to discuss about if rich families would be more probably to become extinct than poor ones. But we would answer that it does not depend on the number but on the distribution and their habitat, as well as the degree of specificity in the resources used by their species. We have not considered whether the families of plants and families of insects correspond to a comparable degree or not, because it is not the aim of our work to discuss these types of questions. Without minimizing this question, the question we are trying to answer is another: to what extent did insects and angiosperms evolve at the same time?

The last question: "Why would one expect family level diversity in plants to have any effect on family level diversity in insects: what assumptions are built into that?" is more like the ones revealed in this manuscript. We do not analyze why this relationship occurs or does not occur, but rather we try to demonstrate if it really occurs, as it has been stated in different works over time without having offered any empirical proof that this is really the case.

7. Another caveat is that the PyRate models assume that rates are the same at any moment in time in all lineages. Of course we know that taxon-specific rates exist in insects. What are the consequences of

this breakage of assumptions for accurately estimating rates? I know the methods are what we have, but the reader needs to be alerted to caveats.

We agree with this comment. Indeed, the PyRate birth-death models assume that rates are homogeneous across clades but heterogeneous over time. We are also aware that rates are likely to have varied across the phylogeny of insects (e.g. Condamine *et al.* 2016 – *Sci. Rep.*; Blaimer *et al.* 2023 – *Nat. Comm.*; Kawahara *et al.* 2023 – *Nat. Ecol. Evol.*). During the revision, we have performed analyses with insect pollinator families only to study their specific diversification. All these results are presented in the revised version of the manuscript. Nonetheless, we have also added a section with a few sentences to present the limits of the methods used in this study.

8. If you keep the existing text and figures, I have minor suggested edits:

- i) Line 15 drivers
- ii) Line 47 delete and
- iii) Line 77 using angiosperm resources extensively until the
- iv) Line 131: dual influence?
- v) Line 134: Nevertheless, the rise of
- vi) Line 136: correlated with reduced origination rates
- vii) Line 145: focussed on the
- viii) Figure 3 legend and axis labels: No units or title on the x-axis scale: from the main text one would naively assume this to be a correlation coefficient but it cannot be. Why not have the confidence intervals on the bars?
- ix) Line 172: cell sizes; herbaceousness,
- x) Line 185: approach that
- xi) I felt Figure 4 was not needed; it's not results and this is not a review paper.
- xii) Line 260: found that flowering plant diversity correlates with a faster
- xiii) Line 266-7. Other hand, a link seems to exist between
- xiv) Line 335: need to say what taxonomic level these data exist at.
- xv) Line 367. I didn't have access to the data, but they certainly weren't in the main text: supplementary materials?

Thank you for all these suggestions. All these minor mistakes were considered and many of them were modified accordingly.

Figure 4 was initially included because the motivation of the paper was to be a review. However, after the comments from the editor, the paper is more and more a new-results manuscript. With that, we agree with the reviewer and because we need more space for the new figures from the analysis, we decided to delete Figure 4.

Reviewer #3 (Remarks to the Author):

This manuscript investigates the potential effects of flowering plants on insect diversification. The link between angiosperms and insects has been hypothesized and studied for quite some time, prompted by the many ecological relationships through pollination and other interactions. Given the high diversity of insects, it is natural that this is a topic of widespread interest.

We thank you for your review and comments mostly associated to the methodological aspects of our study. We have tried to address all of them.

The methods calculate origination, extinction, and diversity from the fossil record, but the techniques do not appear to account for sampling biases or methodological shortcomings of range-through diversity methods. Most notable is the “pull of the recent,” which is visible in the family accumulation plot in figure 2. When information from the extant insect fauna is combined with fossil data, this has the effect of inflating diversity and reducing origination and extinction in more recent time periods. (Technically, it is older time periods that have artificially-low diversity because they do not benefit from the range extensions caused by including extant data.)

We respectfully disagree that the method cannot account for the known biases of the fossil record. PyRate is a Bayesian process-based model that is specifically designed to simultaneously model the rates of preservation and their variation through time and across taxa, the times of origin and extinction of each taxon, and then the rates of origination and extinction through time. By estimating the rates of preservation, PyRate “corrects” the ages of origin and extinction of each taxon instead of directly counting the taxa as they appear in the fossil record. In this way, we move from observed ages to estimated ages, which can alleviate some of the problems of the fossil record, such as artificial peaks in extinction and speciation due to lack of sampling or the reservoir effect. It is important to mention that PyRate has been thoroughly tested under a wide range of conditions, such as low levels of preservation (down to 1–3 fossil occurrences per species on average), severely incomplete taxon sampling (up to 80% missing), and high proportion of singletons (exceeding 30% of the taxa in some cases) (Silvestro *et al.* 2014 – *Syst. Biol.*, 2015 - *PNAS*, 2019 – *Paleobiol.*). As opposed to other methods (including boundary-crossers and three-timers), which are prone to edge effects and tend to flatten the extinction estimates, especially during mass extinctions, PyRate recovers the dynamics of speciation and extinction rates, including sudden rate changes and mass extinctions (Silvestro *et al.* 2019 – *Paleobiol.*).

Because the analysis didn’t account for specific challenges of the fossil record, many of the conclusions about the role of angiosperms could instead be spurious correlations between the angiosperm dominance time series and the pull of the recent. Likewise, reduced origination during times of high diversity (lines 135-137) could be diversity dependence, but also could reflect the pull of the recent bias. It is possible that there are true effects, but I don’t think the argument is convincing now because the methods are not appropriate for dealing with fossil data, and do not attempt to discuss or address widely-known biases.

We also respectfully disagree that the method is inappropriate for dealing with fossil data, and that we have not accounted for the known biases of the fossil record. We have responded to the latter point above. We have added a dedicated section to discuss widely known biases and to show the limitations of our study. Regarding spurious correlations, the Bayesian framework of PyRate also allows multiple hypotheses about the drivers of diversification to be tested simultaneously (Lehtonen *et al.* 2017 - *Sci. Rep.*). The multivariate birth-death model we use has been shown to be robust to several biases. Silvestro *et al.* (2017 - *Evol. Ecol. Res.*) show that the model, and in particular its new implementation based on the horseshoe prior (which we used), can robustly infer diversity dependence within clades from fossil data in a wide range of diversification scenarios. However, in the limitations of the study, we explained that our work did not explore many variables and that further studies are needed. We also recognize that our study provides testable hypotheses for future studies that may challenge and/or complement our results as new data and models become available.

A lot of the discussion section seemed to be on topics that are tangential to the results, rather than making the case that the results are true biological patterns, or explaining why the drivers had an effect on insect diversification. For example, why might spore plants also have influenced insect diversification? A lot of the discussion reads like background information about angiosperm evolution, which is interesting, but I didn't think it was strongly connected to your specific goal in this manuscript. We agree with the reviewer. Because of that we performed new analyses and changed the focus of the discussion, deleting information about evolution of angiosperms and including new references, also following the instructions from the reviewer 2. We hope the new version satisfies the reviewer now.

In general, I think it is extremely difficult to work with origination and extinction in the insect fossil record (even harder than trying to reconstruct diversity, which itself is very challenging). Shifts in the dominant preservation mode (amber vs. compression fossils) from one time period to the next cause apparent origination/extinction spikes because the two modes tend to record different types of families. There are huge variations in the number of insects recorded in different time periods, including very few large localities between about 100 Ma and 50 Ma. Because of that, a lot of extinctions that actually occurred later in the Cretaceous will all cluster earlier. Originations will also be artificially clustered at some of the "super-Lagerstätte" (the extraordinarily well-sampled Baltic amber is one example; likely a number of those families evolved earlier but have not yet been discovered because of the limited Late Cretaceous and Paleocene record). It may be possible to extract some real signal, but I'm fairly skeptical, and it would require careful analysis that deals with the complexity of the insect fossil record.

We do agree with this general comment about the difficulty of estimating reliable diversification rates from the fossil record. The fossil record is inherently incomplete and heterogeneous across clades and through time, making any estimates of diversification processes tentative. Accordingly, we must take these sampling biases into account. This is precisely the aim of PyRate models, which attempt to model simultaneously the rates of preservation and their variation across taxa, the times of origination and extinction of each taxon, and then the rates of origination and extinction through time. By estimating the preservation rates, PyRate corrects the ages of origination and extinction of each taxon. We thus move from observed ages to estimated ages, which can alleviate some issues of the fossil record, such as the artificial peaks of extinction and speciation due to no sampling or Lagerstätte effect, respectively. PyRate is not the final solution to the problems found when studying the fossil record, but it's a step forward. In the revised version, we have added a section that presents and discusses the limitations of our study, including the problems with the fossil record that you mention here.

It was also difficult to evaluate the results because I couldn't find table 1 and there didn't appear to be a dataset containing the potential drivers (spore plant, gymnosperm, angiosperm dominance, temperature, continental fragmentation). Perhaps I missed those on the website, but it would have been very helpful to see a graph of angiosperm dominance through time so that I could visually compare it to insect diversification.

We are sorry about the missing Table 1. It resulted from a misplaced issue at some point of the submission process, which we did different times for some reasons. It should be fixed now. Note that we have now added Table 2 that reports the results obtained for insect pollinator families only. The angiosperm relative diversity has now been added to Figure 2.

Sincerely,
Matthew Clapham

Reviewers' Comments:

Reviewer #1:

Remarks to the Author:

I enjoyed reading the revised version of this manuscript and feel it is a significant improvement upon the last version. I especially liked the addition of new analyses focussed on pollinator families and five major insect orders, as well as the new paragraph on Limitations of this study. While I feel that the authors have adequately responded to most of the comments made by the three reviewers, there is one notable exception that I feel would deserve better revision in the main text (see below). Other than that and a number of minor points noted below, I think this new study will represent an important new contribution to our understanding of the impact that angiosperm diversification had on insect diversification throughout the Cretaceous and Cenozoic.

Family-level analysis of the insect fossil record: both Reviewer #2 and I raised concerns with this approach and questioned its relevance to understanding the background species-level diversification patterns that most readers would ultimately want to learn about. While I understand some of the reasons outlined in the response, it looks as if no modifications were made in the manuscript to justify this approach and consider its potential limitations. Because this is so central to interpretation of this new study, I really think this point would deserve further justification and discussion in the core manuscript so that all readers fully understand the key issue. My suggestion would be to address this in the new paragraph on Limitations of this study. This would not reduce the significance or impact of this work.

Genus-level, relative diversity background data on plants: I would have similar comments on how this point has been addressed in the revision and would have preferred to see the reader provided with more transparent information about the background data, particularly the different taxonomic scale used.

L66-68: "It is thought that the first seed plants were wind-pollinated until some insects diversified and began to feed on gymnosperm ovule secretions in a surface-fluid-feeding manner or on gymnosperm pollen [26–27]." While this has certainly been the default assumption for a long time, I am not so sure everyone would agree now. As shown by numerous paleo papers and recent work by the first author as well as Asar et al., insect pollination among early seed plants may have been so prevalent that it is possible that wind pollination was never ancestral in seed plants as a whole (though crown seed plants are likely to have evolved from ancestors with wind-dispersed spores). It remains of course an open question, but perhaps this sentence could be revised to acknowledge this shifting paradigm?

L123: reword "can have complementary impacted".

L124: reword "We thus, in addition to the relative diversity of angiosperms, we incorporated".

L125: replace "family" with "families".

L251-252: reword "Insects and flowering plants are so obviously made for each other" (sounds too colloquial).

L262: add "orders" after "in only selected insect".

L272-275: "Growing evidence from molecular dated phylogenetics, the fossil record of pollinator insects, palaeontological data on plant morphological characters, and modelling of diversification dynamics, supports the hypothesis that angiosperms evolved in the Early Cretaceous, during a period of peak in insect diversity [20–21, 68]" I think "evolved in the Early Cretaceous" is misleading because we really do not know how old crown angiosperms are. Replace with "diversified significantly" or something similar to acknowledge the possibility that some (but likely not too many) distinct crown

angiosperm lineages might have already existed in the Jurassic?

L285-286: "Angiosperm extinction rates decreased after the K/Pg boundary in parallel with increased speciation [6, 69]" But see the recent study by Thompson and Ramirez-Barahona (<https://royalsocietypublishing.org/doi/10.1098/rsbl.2023.0314>) suggesting a different signal.

L301-302: "Divergent ideas claiming that the advantages of early gymnosperm pollinators did not prevent the decline of gymnosperms [59] should be rejected." I had a hard time understanding this sentence.

L378-380: "In contrast, most angiosperm families (58–80%) originated between ~100 and 90 Ma, during the warmest phases of the Cretaceous [91]." Wrong citation, should be ref. 78.

L381-383: "Despite their age of origin, the rise to ecological dominance of modern angiosperms was geographically heterogeneous and took place over a long period lasting into the Cenozoic, coinciding with the onset of crown diversification in most families [74, 91]." Same comment as above (replace ref. 91 with ref. 78).

Hervé Sauquet

Reviewer #3:

Remarks to the Author:

This is a review of a resubmitted manuscript, so I will focus just on the main issue in my mind, which is whether the study's methods can accurately reconstruct insect diversification at the level of detail required for this analysis. I don't think they can. This is partly due to problems with range-through data used by pyrate (the Pull of the Recent bias), but the main issue isn't even unique to pyrate. The insect record is so challenging that no method is able to reconstruct high-resolution details, especially of origination and extinction, only the broadest strokes of their history.

Observed patterns of diversity have a similar identifiability issue to phylogenies – the same diversity pattern can be produced by multiple combinations of origination, extinction, and preservation. A likelihood-based framework will assign the highest likelihood to a diversification history that resembles the raw data. Unfortunately, the raw diversification pattern of insects almost certainly isn't their true history of diversity, extinction, or origination. Has pyrate been tested on simulated datasets where the true diversification history differs significantly from the observed data? Even though pyrate is a sophisticated method, I can't see how it would be able to identify the true diversification as the most likely result unless the true diversification is basically the same as the raw data.

For all of pyrate's correcting for preservation, the diversity curve in figure 2 is nearly identical to the raw diversity curve, with some minor smoothing and a small error envelope. There are three sharp jumps in diversity, all coinciding with exceptional deposits (Daohugou and Karatau in the mid/late Jurassic; Yixian, Crato, Baissa, etc. in the Early Cretaceous; Eocene ambers, Florissant, Green River, etc. in the early Cenozoic). This raises concerns about the accuracy of extinction and origination rates. It seems like a very big coincidence if true increases in diversity always happened to align with super-Lagerstatten!

The diversity pattern in figure 2 also isn't consistent with the description of insect diversity in the introduction (e.g. lines 54-56). The introduction states that insect diversity had a peak in the Early Cretaceous but figure 2 suggests a 50% increase in diversity between the Early Cretaceous and late Cenozoic.

It also doesn't appear that pyrate accounts for the Pull of the Recent in any way. (Incidentally, the Pull

of the Recent is the reason why figure 2 doesn't agree with the statement in the introduction.) For groups where this bias is strong, such as insects, the Pull of the Recent will lead to a spurious decrease in extinction rates towards the present. This will artificially inflate the strength of the relationship between angiosperms and extinction, for example. When comparing previous works that tried to reconstruct insect diversity, the biggest difference is not in the method itself (pyrate, capture-recapture, etc.), but whether it is affected by the Pull of the Recent, which is an enormous bias in the insect record.

The paper doesn't provide a figure showing origination and extinction rates, which would be very interesting. I suspect that there is an extinction peak in the mid-Cretaceous (around the Albian-Cenomanian), which again would show whether pyrate can adequately correct for preservation. Although this turnover has been discussed in the literature, it appears much more abrupt in the raw data than it would have been in reality. The abruptness of extinction is driven by the shift from dominantly-compression to dominantly-amber fossilization in the mid-Cretaceous, coupled with the near-absence of compression fossil localities between the Cenomanian and the late Paleocene. The mid-Cretaceous extinction peak reflects ~50 million years of extinction mostly compressed into one time interval because of taphonomy and sampling.

However, even though we can be confident that the mid-Cretaceous extinction peak is inflated by preservation-related range truncations (in fact, very likely didn't exist as a distinct extinction peak), that doesn't mean we can reconstruct the true pattern. For example, I wouldn't be surprised if there was some kind of insect extinction at the K/Pg, but the Maastrichtian record is so exceptionally poor that it is impossible to tell.

The response to reviewers states repeatedly that pyrate can account for sampling, determine the true origination and extinction times, etc. – however, no method is a panacea. Just because something works in some circumstances, doesn't mean that it will work in all situations. Examination of the reconstructed diversity history in figure 2, in combination with a knowledge of the insect fossil record, reveals a number of red flags that suggest that pyrate is basically returning the raw data and isn't a magical "silver bullet" that can overcome all problems.

Sincerely,

Matthew Clapham

Reviewer #4:
Remarks to the Author:
Dear Editor,

I have now read the revised version of the manuscript "The dual role of the angiosperm radiation in the global diversification of insects and insect pollinators" by Peris and Condamine, as well as the three reviewer reports from the previous round.

Like the previous reviewers, I consider this to be a very interesting study that addresses a fundamental question in evolutionary biology, and I agree that the results should be appealing to a wide audience. The new version appears to have improved analyses and the logic of the study is now better. Especially performing the analyses separately for all insects and then pollinators, and then separately for five main insect orders provides added insight into the focal questions. The use of PyRate its Bayesian MBD approach seems like an excellent solution for this type of a study and, as suggested by the previous reviewers, the limitations of the method are now discussed in the Discussion.

Despite the improvements, I find that some issues still remain, and that some relevant suggestions by the reviewers of the previous round have not been entirely followed in the preparation of the new version of the manuscript. I have divided my comments below to major and minor comments, of which the latter mainly concern small typos and unclear sentences.

MAJOR COMMENTS

1. Slightly contrasting the comments by the previous reviewers, I think that the solution to use family-level diversity estimates for insects is justifiable for a fossil-based analysis, and the same applies to the use of genus-level estimates for plants. The use of different measures for plants and insects does not seem to be a problem as such, but in both cases, the solution to use higher-level taxa should be described and justified briefly in the text.

2. The fact that plant diversity is measured at the level of genera is still not mentioned in the text (see comment 3(a) by Reviewer 1 of the previous round).

3. In a response to comment 3(b) by Reviewer 1 concerning L140-141 of the original manuscript, the authors note that they «could not assess the correlation between angiosperms and gymnosperms in this study with the data at hand». This seems odd, as the correlation can be checked using only relative values, and that the relative values most likely have been obtained from some estimate of absolute numbers of genera. The article of Silvestro et al. (2015) mainly shows rates in origination and extinction, and I could not find data on absolute or relative plant diversities in their article or its Supplementary information. Some pieces of information are apparently missing from the description on how the data on plant diversity was obtained for the present study.

4. The addition of a section on analytical limitations is very good. It might be worth mentioning how the fossil-based approach and PyRad differs from phylogeny-based approaches based on extant species only, and how different approaches might provide complementary insights. Furthermore, it would be advisable to briefly discuss the fact that the definition of families (and genera in the case of plants) is not entirely unambiguous, and how these diversity measures are (or are not) related to species-level diversity. Also, the effect of using relative genus-level diversities for plants should be discussed, as this will automatically create partial interdependence across the three plant groups - how may this affect the analyses and results?

5. As pointed out by Reviewer 1, the statement of a: "Peak in insect diversity" during the Cretaceous (L20-22, L53-56, L275, L376-377) is a highly unclear formulation in this context. Yes, there is technically a peak preceding a temporary decrease (Fig. 2), but current diversity is far higher. Somehow, this should be expressed in a clearer way in the sentences in question, by mentioning that the peak was transient.

6. All three reviewers from the previous round point out that the Discussion has a lot of text on the drivers of diversification in angiosperms, which was not the focus of this study. This issue has been partly alleviated by the changes made in the new version, but there is still unnecessarily much text on plant diversification in the middle of the Discussion (e.g., L263-268, L285-298). I would suggest putting most text on plant diversification to a new first or second paragraph of the Discussion (right after an opening paragraph), together with a brief description of climatic and tectonic/continental changes through time that could have influenced speciation or extinction rates in insects. After that, the text should focus mainly on the effects of plant diversity and the other factors (including density dependence with regard to insect family richness) on insect diversification (and the various partitions of insect taxa and ecological groups). Here, I would refer to comment 5 by Reviewer 2 of the previous round.

7. L318-326: These rows basically repeat text from the Results, without putting the analyzed factors

into a broader context by relating the results to previous studies. One thing that should be discussed is that a large fraction of insect families are neither pollinators nor herbivores. For example, most families of Hymenoptera are parasitoids that interact with plants only indirectly, and similar situations may be present in other insect taxa. Therefore, some mention should be made of why we would expect angiosperm diversity to influence speciation or extinction rates in such groups.

8. Tables 1-3: Some shrinkage weights over 0.5 for which the 95% CI overlaps with 0.5 are in bold – is this correct? Table 2: Some correlation coefficients for which the 95% CI overlaps with 0 are in bold – is this correct? Seems like the boldings in the tables need to be checked carefully, and/or that the legends need to be clarified.

9. It is unclear from the materials in the review package whether the raw data and scripts used in the analyses will be deposited somewhere.

MINOR COMMENTS

1. L18-19: First sentence of Abstract is unclear.
2. L45: Unclear.
3. L49-50: Odd formulation, b/c the replacement is the turnover.
4. L78-79: Unclear.
5. L98: Cretaceous?
6. L100: "increase in terrestrial biodiversity" seems unnecessarily broad in this context.
7. L109: "a"  "the"?
8. L113: "that"  ", which"?
9. L113: "alleviate some issues" -- need to be more specific here.
10. L117: bibliography?
11. L125: family  families
12. L126-127: Temperature, not change in temperature?
13. L130: "using"  "estimated using"?
14. L130-131: This requires clarification.
15. L132: extinction  extinctions
16. L132: "Period of increase in terrestrial biodiversity" is too broad when the text is only about plants.
17. L141-144: This should be in the Discussion, not in the Results.
18. L146: The indices could be described/defined very briefly at first mention.

19. Figure 3 and 4: Some of the columns have been cut, and this should be indicated in the columns somehow.
20. L137-191: Consider listing correlation coefficients first and then shrinkage values, to correspond to the order in the tables in the Supplementary information.
21. L185-187: Can the correlation values be compared like this directly, without considering the 95% HPDs in Table S2?
22. L187-191: Highly unclear sentence that feels like it belongs in the Discussion.
23. L195-196: "Order-specific families"?
24. L222: "It results"?
25. L224-225, L233-234: Feels like Discussion?
26. L238-240: To Discussion?
27. L240: "gymnosperm diversity"  "relative gymnosperm diversity"
28. L242-244: This is pure Discussion, not Results. Also, because the additional factors are included in the analyses, they should be discussed.
29. L247: groups  group
30. L247-255: The hypothesized role of angiosperms for herbivore diversification should also be mentioned. There are numerous studies focusing on this.
31. L258: Unclear what "evolution of herbivory following cycles of host shifts" means.
32. L283-284: This seems like a very strong assumption?
33. L327-337: The end of the main Discussion focuses mainly on plant diversification, which was not the focus of the main analyses.
34. L348: "biases"
35. L398-399: Odd sentence to end the Conclusions?
36. L411-412: Unclear.
37. L413: "PyRate has developed"?
38. L445-473: Since the emphasis of the study is on the effects of changes in relative diversity of different plant groups, it would seem more logical to first list plant-related variables and then the others.
39. L467: If plant diversity is measured as relative numbers of genera, it should be mentioned here.
40. L481: Promoted?
41. The relevant comment 8(viii) of Reviewer 2 has not been responded to, and x-axis units are still missing from Figs. 3 and 4.

RESPONSE TO REVIEWERS' COMMENTS

Reviewer #1 (Remarks to the Author):

I enjoyed reading the revised version of this manuscript and feel it is a significant improvement upon the last version. I especially liked the addition of new analyses focussed on pollinator families and five major insect orders, as well as the new paragraph on Limitations of this study. While I feel that the authors have adequately responded to most of the comments made by the three reviewers, there is one notable exception that I feel would deserve better revision in the main text (see below). Other than that and a number of minor points noted below, I think this new study will represent an important new contribution to our understanding of the impact that angiosperm diversification had on insect diversification throughout the Cretaceous and Cenozoic.

We are pleased the reviewer likes our revised version of the manuscript. We have addressed the last major comment and incorporated all the minor ones.

Family-level analysis of the insect fossil record: both Reviewer #2 and I raised concerns with this approach and questioned its relevance to understanding the background species-level diversification patterns that most readers would ultimately want to learn about. While I understand some of the reasons outlined in the response, it looks as if no modifications were made in the manuscript to justify this approach and consider its potential limitations. Because this is so central to interpretation of this new study, I really think this point would deserve further justification and discussion in the core manuscript so that all readers fully understand the key issue. My suggestion would be to address this in the new paragraph on Limitations of this study. This would not reduce the significance or impact of this work.

Genus-level, relative diversity background data on plants: I would have similar comments on how this point has been addressed in the revision and would have preferred to see the reader provided with more transparent information about the background data, particularly the different taxonomic scale used.

Thank you very much for the clarification. We agree with this concern but we did not understand from the first review that this information was required to appear in the manuscript itself. As the reviewer suggested, we have now included a new paragraph on “*Limitations of the study*”, including an explanation about the use and limits of family-level dataset for insects compared with genus-level dataset for plants, and we tentatively explain why the different taxonomic scales do not drastically alter our results. We have also explained how the plant relative diversity has been computed from the results of Silvestro et al. (2015 – *New Phytol.*) in the section *Methods* of the revised manuscript.

L66-68: “It is thought that the first seed plants were wind-pollinated until some insects diversified and began to feed on gymnosperm ovule secretions in a surface-fluid-feeding manner or on gymnosperm pollen [26–27].” While this has certainly been the default assumption for a long time, I am not so sure everyone would agree now. As shown by numerous paleo papers and recent work by the first author as well as Asar et al., insect pollination among early seed plants may have been so prevalent that it is possible that wind pollination was never ancestral in seed plants as a whole (though crown seed plants are likely to have evolved from ancestors with wind-dispersed spores). It remains of course an open question, but perhaps this sentence could be revised to acknowledge this shifting paradigm?

Thank you for the suggestion. We completely agree with this point. It was our intention to shift this paradigm with the sentences that followed the lines highlighted by the reviewer. In order to be clearer with this objective we changed the verbal tense: It is thought → It was thought.

L123: reword “can have complementary impacted”.
It has now been changed to: “have also complementary impacted”.

L124: reword “We thus, in addition to the relative diversity of angiosperms, we incorporated”.
We corrected the sentence by deleting the second “we”.

L125: replace “family” with “families”.
It has been replaced.

L251-252: reword “Insects and flowering plants are so obviously made for each other” (sounds too colloquial).
The sentence has been deleted from the text.

L262: add “orders” after “in only selected insect”.
It has been added.

L272-275: “Growing evidence from molecular dated phylogenetics, the fossil record of pollinator insects, palaeontological data on plant morphological characters, and modelling of diversification dynamics, supports the hypothesis that angiosperms evolved in the Early Cretaceous, during a period of peak in insect diversity [20–21, 68]” I think “evolved in the Early Cretaceous” is misleading because we really do not know how old crown angiosperms are. Replace with “diversified significantly” or something similar to acknowledge the possibility that some (but likely not too many) distinct crown angiosperm lineages might have already existed in the Jurassic?
We agree with the reviewer. Thank you for the suggestion. The sentence was reworded accordingly.

L285-286: “Angiosperm extinction rates decreased after the K/Pg boundary in parallel with increased speciation [6, 69]” But see the recent study by Thompson and Ramirez-Barahona (<https://royalsocietypublishing.org/doi/10.1098/rsbl.2023.0314>) suggesting a different signal.
Thank you for pointing this new study. This newly detected effect has been included in the list of references and added in the text.

L301-302: “Divergent ideas claiming that the advantages of early gymnosperm pollinators did not prevent the decline of gymnosperms [59] should be rejected.” I had a hard time understanding this sentence.
Thank you for the note. We have rephrased this sentence and we hope it is clearer now.

L378-380: “In contrast, most angiosperm families (58–80%) originated between ~100 and 90 Ma, during the warmest phases of the Cretaceous [91].” Wrong citation, should be ref. 78.
Thank you. Ref. 78 has been cited in this place instead of ref. 91.

L381-383: “Despite their age of origin, the rise to ecological dominance of modern angiosperms was geographically heterogeneous and took place over a long period lasting into the Cenozoic, coinciding with the onset of crown diversification in most families [74, 91].” Same comment as above (replace ref. 91 with ref. 78).

The reference 91 was replaced by the reference 78 in both cases.

Hervé Sauquet

Reviewer #3 (Remarks to the Author):

This is a review of a resubmitted manuscript, so I will focus just on the main issue in my mind, which is whether the study's methods can accurately reconstruct insect diversification at the level of detail required for this analysis. I don't think they can. This is partly due to problems with range-through data used by pyrate (the Pull of the Recent bias), but the main issue isn't even unique to pyrate. The insect record is so challenging that no method is able to reconstruct high-resolution details, especially of origination and extinction, only the broadest strokes of their history.

Thank you for reviewing again our study. We appreciate and understand the reviewer's comment, and we feel very sorry that we fail to convince him that PyRate can handle some known biases pertaining to the nature of the insect fossil record. Here, we would like to remember that we aim at studying a broad stroke of the evolution of insects and angiosperms. We also want to remind that we do not intend to show high resolution details of insect diversification. The conclusions are broad enough to present general tendencies and patterns, which provide new testable hypotheses for future studies that, we hope, will come with better data and new models.

Observed patterns of diversity have a similar identifiability issue to phylogenies – the same diversity pattern can be produced by multiple combinations of origination, extinction, and preservation. A likelihood-based framework will assign the highest likelihood to a diversification history that resembles the raw data. Unfortunately, the raw diversification pattern of insects almost certainly isn't their true history of diversity, extinction, or origination. Has pyrate been tested on simulated datasets where the true diversification history differs significantly from the observed data? Even though pyrate is a sophisticated method, I can't see how it would be able to identify the true diversification as the most likely result unless the true diversification is basically the same as the raw data.

To our knowledge, there has been no formal demonstration that Bayesian fossil-based process-based inferences of diversification with PyRate have identifiability issues. The identifiability issues in birth-death models fitted to molecular phylogenies only apply to a specific class of birth-death models that assume homogeneous rates across clades in a phylogeny (Louca & Pennell 2020 – Nature). Even in this case, there are solutions to circumvent these issues (see Helmstetter et al. 2022 – Syst. Biol.; Morlon et al. 2022 – TREE).

PyRate has been thoroughly tested with many simulation schemes (see Silvestro et al. 2014 – Syst. Biol.; Silvestro et al. 2015 – PNAS; Silvestro et al. 2019 – Paleobiology) such as low levels of preservation (down to 1–3 fossil occurrences per taxon on average), severely incomplete taxon sampling (up to 80% missing), and high proportion of singletons (exceeding 30% of the taxa in some cases). As opposed to other

methods (including boundary-crossers and three-timers), which are prone to edge effects and tend to flatten the extinction estimates, especially during mass extinctions, PyRate recovers the dynamics of speciation and extinction rates, including sudden rate changes and mass extinctions (Silvestro et al. 2019 – Paleobiology). Importantly, the Bayesian framework of PyRate also allows testing hypotheses related to the drivers of diversification (Silvestro et al. 2015 – PNAS; Lehtonen et al. 2017 – Sci. Rep.; Haggen et al. 2018 – Syst. Biol.).

In addition, the information is contradictory between reviewers when reviewer #4 says: The use of the Bayesian PyRate MBD approach seems like an excellent solution for this type of a study. As suggested by the reviewers, the limitations of the method are now discussed in the Discussion.

For all of pyrate's correcting for preservation, the diversity curve in figure 2 is nearly identical to the raw diversity curve, with some minor smoothing and a small error envelope. There are three sharp jumps in diversity, all coinciding with exceptional deposits (Daohugou and Karatau in the mid/late Jurassic; Yixian, Crato, Baissa, etc. in the Early Cretaceous; Eocene ambers, Florissant, Green River, etc. in the early Cenozoic). This raises concerns about the accuracy of extinction and origination rates. It seems like a very big coincidence if true increases in diversity always happened to align with super-Lagerstätte!

We understand and agree with the comment on the impact of Lagerstätte effect on rates through time, which is a well-known issue in the fossil record. The fossil record of insects makes no exception to this bias, which is especially true given the long evolutionary history of the group. For some groups, such as Ephemeroptera (Sroka et al. 2023 – Sci. Rep.) and Plecoptera (Jouault et al. 2022 – Insect Syst. Div.), this is particularly obvious. In such extreme cases, rates of origination and extinction are likely inaccurate, but they can also reflect a biological process (e.g. changes in ecological habits after a sudden extinction that can alter the preservation mode). However, at the scale of all insects, this effect is more diluted (Condamine et al. 2016 – Sci. Rep.) probably because insects comprise a broad range of ecological habits with varying preservation modes.

To address the reviewer's comment, we estimated the origination and extinction rates using the Bayesian birth-death model with constrained shifts (rates being constant in bins, with bins being the geological epochs); the same model as in Condamine et al. (2016 – Sci. Rep.). Regarding the exceptional deposits (Middle-Late Jurassic, Early Cretaceous, and Eocene), we found that origination and extinction rates do not stand out as outliers in the overall diversification dynamics, despite the jumps in diversity. It seems that origination and extinction rates are both more elevated during the Paleozoic than elsewhere. We think these results need to be shown and discussed in the main text. Accordingly, the results are presented as Supplementary Figure 1 and reported in the main text (section *Limitations of the study*) to discuss the issue of preservation that can distort estimation of diversification rates.

The diversity pattern in figure 2 also isn't consistent with the description of insect diversity in the introduction (e.g. lines 54-56). The introduction states that insect diversity had a peak in the Early Cretaceous but figure 2 suggests a 50% increase in diversity between the Early Cretaceous and late Cenozoic.

We think that the peak in insect diversity during the Early Cretaceous is clearly observed in Figure 2. We agree that insect diversity increased again during the Cenozoic, much more than during the Cretaceous, which has been associated to the angiosperm effect after our analyses (in addition to other complementary causes). We find it strange that the

reviewer does not accept this diversification peak when it has been cited in previous manuscripts, some of which he has collaborated on (e.g., Ref. 20: Clapham et al. 2016 – PRSB). Nevertheless, we used the suggestion from reviewer #4 and included “transient” in the description of the cited peak. We hope this explanation satisfies the reviewer.

It also doesn't appear that pyrate accounts for the Pull of the Recent in any way. (Incidentally, the Pull of the Recent is the reason why figure 2 doesn't agree with the statement in the introduction.) For groups where this bias is strong, such as insects, the Pull of the Recent will lead to a spurious decrease in extinction rates towards the present. This will artificially inflate the strength of the relationship between angiosperms and extinction, for example. When comparing previous works that tried to reconstruct insect diversity, the biggest difference is not in the method itself (pyrate, capture-recapture, etc.), but whether it is affected by the Pull of the Recent, which is an enormous bias in the insect record.

The pull of the recent describes a phenomenon in which a combination of factors causes paleontologists to overestimate diversity into the present. Biased preservation and sampling in the fossil record led to lower estimates of past diversity, with modern taxa being considered more diverse because present diversity is the best sampled (Raup 1979 - *Bul. Carnegie Mus. Nat. Hist.*). However, the overall effect of the pull of the recent does not appear to be as large as originally thought (Raup 1972 - *Science*; Sahney & Benton 2017 - *Evol. Ecol. Res.*). While there are undoubtedly gaps in the fossil record, there is no reason to believe that the pull of the recent has significantly affected the observed paleodiversity patterns of insects. There is no clear pull of the recent in the insect family's dataset. This means that the great expansion of diversity over the past 120 Myrs is reasonably accurate (Benton & Storrs 1994 – *Geology*; Sahney et al. 2010 – *Biol. Lett.*; Kalmar & Currie 2010 – *Paleobiol.*). At the insect family level, it's not that crazy that extinction rate decreased toward the present because of the taxonomic level used, but we could not say so at the species level or genus level. However, working at lower taxonomic levels is still in its infancy (see Condamine et al. 2020 – *Cladistics*; Jouault et al. 2022 – *ISD*; Jouault et al. 2022 – *Nat. Comm.*). PyRate accounts for the number of extant taxa in the clade with a prior (-N option in PyRate) to correct for the pull of the recent. Anyway, we agree that no method is immune to the effect of the pull of the recent, but all studies have this issue, so we still think we can propose a study on the macroevolution of insects if we have explained the hypotheses and the model limitations, which we have done in the *Limitations* section.

The paper doesn't provide a figure showing origination and extinction rates, which would be very interesting. I suspect that there is an extinction peak in the mid-Cretaceous (around the Albian-Cenomanian), which again would show whether pyrate can adequately correct for preservation. Although this turnover has been discussed in the literature, it appears much more abrupt in the raw data than it would have been in reality. The abruptness of extinction is driven by the shift from dominantly-compression to dominantly-amber fossilization in the mid-Cretaceous, coupled with the near-absence of compression fossil localities between the Cenomanian and the late Paleocene. The mid-Cretaceous extinction peak reflects ~50 million years of extinction mostly compressed into one time interval because of taphonomy and sampling.

As discussed above regarding the Lagerstätte effect on rates through time, we have addressed the reviewer's comment with the estimation of origination and extinction rates through time for all insects. Relying on the Bayesian birth-death model with constrained shifts (rates being constant in bins, with bins being the geological epochs), we did not

find that extinction rates peaked in the Early Cretaceous. On the contrary, extinction rates throughout the Cenozoic and Mesozoic are rather homogeneous and low; they only stand out during the Paleozoic as being more elevated, which could be due to sampling artifacts or the Permian crises (Jouault et al. 2022 – Nat. Comm.). As said above, we report these results as Supplementary Figure 1 and in the main text (section *Limitations of the study*) to discuss the issue of preservation that can distort estimation of diversification rates.

However, even though we can be confident that the mid-Cretaceous extinction peak is inflated by preservation-related range truncations (in fact, very likely didn't exist as a distinct extinction peak), that doesn't mean we can reconstruct the true pattern. For example, I wouldn't be surprised if there was some kind of insect extinction at the K/Pg, but the Maastrichtian record is so exceptionally poor that it is impossible to tell.

We agree that the Maastrichtian fossil record is very poor so that the K/Pg extinction even cannot be assessed thoroughly. However, a sudden and strong extinction does not change the fact that we are here interested in studying the background extinction rates on the long term, i.e. over more than 100 million years. We have added a sentence in the *Limitations* section about the lack of deposits at key periods, but we have also explained that this lack of fossils does not hamper the study of long-term diversification rates.

The response to reviewers states repeatedly that pyrate can account for sampling, determine the true origination and extinction times, etc. – however, no method is a panacea. Just because something works in some circumstances, doesn't mean that it will work in all situations. Examination of the reconstructed diversity history in figure 2, in combination with a knowledge of the insect fossil record, reveals a number of red flags that suggest that pyrate is basically returning the raw data and isn't a magical “silver bullet” that can overcome all problems.

We agree that any model has flaws and cannot tell us the truth. We do not say that PyRate is a perfect model since we recognize the limitations of our study, as for example with the new section entitled with this same name. We also recognize the bias in sampling fossils and their effect in the diversity estimation as previously shown (Ref. 20: Clapham et al. 2016 – PRSB; Ref. 21: Schachat et al. 2019 – PRSB). Our study only proposes an explanation after analysing the insect fossil record with a promising method that has already been successfully applied to different groups to study their deep-time history (e.g. Silvestro et al. 2015 – New Phytol.; Condamine et al. 2019 – PNAS; Condamine et al. 2020 – Cladistics; Jamson et al. 2022 – Palaeontol.; Jouault et al. 2022 – Nat. Comm.; Guo et al. 2023 – Nat. Comm.).

Sincerely,
Matthew Clapham

Reviewer #4 (Remarks to the Author):

Dear Editor,

I have now read the revised version of the manuscript "The dual role of the angiosperm radiation in the global diversification of insects and insect pollinators" by Peris and Condamine, as well as the three reviewer reports from the previous round.

Thank you for reviewing again our study. We appreciate the time and effort you put in this work.

Like the previous reviewers, I consider this to be a very interesting study that addresses a fundamental question in evolutionary biology, and I agree that the results should be appealing to a wide audience. The new version appears to have improved analyses and the logic of the study is now better. Especially performing the analyses separately for all insects and then pollinators, and then separately for five main insect orders provides added insight into the focal questions. The use of PyRate its Bayesian MBD approach seems like an excellent solution for this type of a study and, as suggested by the previous reviewers, the limitations of the method are now discussed in the Discussion.

We are pleased the reviewer likes the revised version of the manuscript and the new results associated with the analyses of the main insect orders. We are also particularly delighted to read that the reviewer finds the PyRate approach appropriate for such a study.

Despite the improvements, I find that some issues still remain, and that some relevant suggestions by the reviewers of the previous round have not been entirely followed in the preparation of the new version of the manuscript. I have divided my comments below to major and minor comments, of which the latter mainly concern small typos and unclear sentences.

Thank you for these additional comments. We have addressed each point, hoping that we have convincingly responded.

MAJOR COMMENTS

1. Slightly contrasting the comments by the previous reviewers, I think that the solution to use family-level diversity estimates for insects is justifiable for a fossil-based analysis, and the same applies to the use if genus-level estimates for plants. The use of a different measures for plants and insects does not seem to be a problem as such, but in both cases, the solution to use higher-level taxa should be described and justified briefly in the text.

Thank you for the suggestion. An explanation has been accordingly included in the section “*Limitations of the study*”, as suggested by reviewer #1. The family level is very relevant, as argued by Labandeira and Sepkoski (1993 – Science) as follows: “*the rank of family was chosen for several reasons. (i) This taxonomic level has been analyzed in other studies of fossil diversity and seems to correlate well with underlying species diversity. (ii) Families are less susceptible to irregular and biased sampling than are fossil species and genera, so that an evolutionary signal is better maintained at this level. (iii) Families of insects, especially extant ones, are reasonably well established through consensus among researchers, whereas fossil species and genera are more idiosyncratically defined and less frequently correspond to good phylogenetic or phenetic units. (iv) Insect families individually possess discrete, often highly stereotyped life habits, and their morphologies directly reflect their trophic guilds, which are informative in diversity studies.*”

2. The fact that plant diversity is measured at the level of genera is still not mentioned in the text (see comment 3(a) by Reviewer 1 of the previous round).

Thank you for the suggestion. We agree that we omitted to mention this key point in our revised version. An explanation has been included in both the section “*Methods*” and the section “*Limitations of the study*”, as suggested by reviewers #1 and #4.

3. In a response to comment 3(b) by Reviewer 1 concerning L140-141 of the original manuscript, the authors note that they «could not assess the correlation between angiosperms and gymnosperms in this study with the data at hand». This seems odd, as the correlation can be checked using only relative values, and that the relative values most

likely have been obtained from some estimate of absolute numbers of genera. The article of Silvestro et al. (2015) mainly shows rates in origination and extinction, and I could not find data on absolute or relative plant diversities in their article or its Supplementary information. Some pieces of information are apparently missing from the description on how the data on plant diversity was obtained for the present study.

Thank you for this comment. Indeed, the study by Silvestro et al. (2015 – *New Phytol.*) only shows rates of origination and extinction for major plant lineages, but not the diversity dynamics through time. In previous studies (e.g. Lehtonen et al. 2017 – *Sci. Rep.*; Condamine et al. 2020 – *PNAS*), the diversity dynamics of these plant groups have been computed based on the results from the study of Silvestro et al. (2015 – *New Phytol.*). Specifically, Silvestro et al. (2015 – *New Phytol.*) estimated the times of origination and extinction of all plant genera in their dataset, which allows computing the temporal dynamic of diversity changes.

4. The addition of a section on analytical limitations is very good. It might be worth mentioning how the fossil-based approach and PyRad differs from phylogeny-based approaches based on extant species only, and how different approaches might provide complementary insights. Furthermore, it would be advisable to briefly discuss the fact that the definition of families (and genera in the case of plants) is not entirely unambiguous, and how these diversity measures are (or are not) related to species-level diversity. Also, the effect of using relative genus-level diversities for plants should be discussed, as this will automatically create partial interdependence across the three plant groups - how may this affect the analyses and results?

We agree with this new comment on the limits of the study. We have expanded the *Limitations* section by including several new aspects mentioned here (different taxonomic levels studied) but also from the two other reviewers (e.g. transient diversity peak in the mid-Cretaceous).

5. As pointed out by Reviewer 1, the statement of a: "Peak in insect diversity" during the Cretaceous (L20-22, L53-56, L275, L376-377) is a highly unclear formulation in this context. Yes, there is technically a peak preceding a temporary decrease (Fig. 2), but current diversity is far higher. Somehow, this should be expressed in a clearer way in the sentences in question, by mentioning that the peak was transient.

The Early Cretaceous peak in insect diversity is not a result from our analyses but has been widely stated in previous works (Ref. 20: Clapham et al. 2016 – *PRSB*; Ref. 21: Schachat et al. 2019 – *PRSB*). We followed the nomenclature used in these previous works because we recovered the "same" effect. However, we agree that we could have been more precise to describe the peak as transient. Thank you for the alternative explanation.

6. All three reviewers from the previous round point out that the Discussion has a lot of text on the drivers of diversification in angiosperms, which was not the focus of this study. This issue has been partly alleviated by the changes made in the new version, but there is still unnecessarily much text on plant diversification in the middle of the Discussion (e.g., L263-268, L285-298). I would suggest putting most text on plant diversification to a new first or second paragraph of the Discussion (right after an opening paragraph), together with a brief description of climatic and tectonic/continental changes through time that could have influenced speciation or extinction rates in insects. After that, the text should focus mainly on the effects of plant diversity and the other factors (including density dependence with regard to insect family richness) on insect diversification (and the

various partitions of insect taxa and ecological groups). Here, I would refer to comment 5 by Reviewer 2 of the previous round.

Thank you for this follow-up comment. We have now deleted the lines 263-268. We have rephrased the lines 285-289 to explain the situation for gymnosperms during the Cretaceous. This latter information is also needed to understand the following sentence: “Faced with this situation, gymnosperm pollinators likely had little options but to adapt or go extinct, depending on how specialized they were”, and subsequent information about the evolution of pollinator insect lineages during the Cretaceous.

As the reviewer says, we deleted from the first version all the possible information about plant diversification and kept only the necessary information to understand the line of the discussion. We have now moved the paragraph from Results to Discussion: “Additional different alternative hypotheses explaining the radiation of insects are also proposed in the literature (e.g., [12, 56]) but are not covered in this analysis, which focuses on the angiosperm-insect co-diversification” following the reviewer suggestion from above. Thus, the proposed ideas by the reviewer in this point are out of the intention with this manuscript, which would imply to completely rewrite a new discussion.

7. L318-326: These rows basically repeat text from the Results, without putting the analyzed factors into a broader context by relating the results to previous studies. One thing that should be discussed is that a large fraction of insect families are neither pollinators nor herbivores. For example, most families of Hymenoptera are parasitoids that interact with plants only indirectly, and similar situations may be present in other insect taxa. Therefore, some mention should be made of why we would expect angiosperm diversity to influence speciation or extinction rates in such groups.

As we explained earlier, it is not our intention to analyse all the factors that influenced insect diversity, but to focus on angiosperms effects. We agreed that parasites and parasitoids are of importance and deserves special mention. Because of that, we added earlier in the discussion the text: “One specific case is the one that refers to parasite and parasitoid insect lineages, because they together are estimated to represent an important amount of the total diversity of extant insects. It has been suggested, however, that the diversification of parasitic and especially parasitoid insect families occurred rapidly during the Late Jurassic–Early Cretaceous and could have been a major driver of the Early Cretaceous peak in family-level insect diversity [21].” Extracted from the reference 21 from already cited (Schachat et al., 2019).

8. Tables 1-3: Some shrinkage weights over 0.5 for which the 95% CI overlaps with 0.5 are in bold – is this correct? Table 2: Some correlation coefficients for which the 95% CI overlaps with 0 are in bold – is this correct? Seems like the boldings in the tables need to be checked carefully, and/or that the legends need to be clarified.

In Tables, the shrinkage weights (ω) greater than 0.5 are highlighted in bold when the 95% CI of the correlation parameters (G) does not overlap with zero. This double criterion indicates whether a correlation is significant or not. The red highlighted lines stand for the correlation with angiosperms, the key hypothesis tested in this study, but is bolded or not depending on the significance threshold explained above. We have indeed bolded some lines by mistake in Table 2 but have now corrected them.

9. It is unclear from the materials in the review package whether the raw data and scripts used in the analyses will be deposited somewhere.

We created a new section in the second version of the manuscript in which we specified this information as follows:

Data availability

All data are originally available in the main text or extracted from [19]. All fossil datasets to repeat the analyses described here are available through the FigShare digital data repository (<https://figshare.com/s/be9b896c854ba619bb05>).

MINOR COMMENTS

1. L18-19: First sentence of Abstract is unclear.

Changed.

2. L45: Unclear.

Changed.

3. L49-50: Odd formulation, b/c the replacement is the turnover.

Changed.

4. L78-79: Unclear.

Changed.

5. L98: Cretaceous?

Deleted.

6. L100: "increase in terrestrial biodiversity" seems unnecessarily broad in this context.

Deleted.

7. L109: "a"  "the"?

Changed.

8. L113: "that"  ", which"?

Changed

9. L113: "alleviate some issues" -- need to be more specific here.

The corresponding sentence has been expanded to be more specific.

10. L117: bibliography?

Deleted.

11. L125: family  families

Changed.

12. L126-127: Temperature, not change in temperature?

Changed.

13. L130: "using"  "estimated using"?

Changed.

14. L130-131: This requires clarification.

Changed.

15. L132: extinction  extinctions

Changed.

16. L132: "Period of increase in terrestrial biodiversity" is too broad when the text is only about plants.

Deleted.

17. L141-144: This should be in the Discussion, not in the Results.

Changed of place.

18. L146: The indices could be described/defined very briefly at first mention.

We agree and we have now described the MBD model and associated parameters in the first place where mentioned as follows: "We then relied on the Bayesian multivariate birth-death (MBD) model implemented in PyRate to simultaneously estimate correlations between diversification dynamics and multiple environmental variables, with the statistical support being estimated with a shrinkage weight (ω) for each correlation parameter (G) for origination ($G\lambda$) and extinction ($G\mu$) depending on each environmental variable [43]."

19. Figure 3 and 4: Some of the columns have been cut, and this should be indicated in the columns somehow.

Changed.

20. L137-191: Consider listing correlation coefficients first and then shrinkage values, to correspond to the order in the tables in the Supplementary information.

The standard presentation is to report first the shrinkage weights to show the strength of the significance and then the correlation parameters for the strength of the correlation with the origination and/or extinction rates. This is what has been done in the studies relying on the MBD analyses (see e.g. Lehtonen et al. 2017 - *Sci. Rep.*; Condamine et al. 2019 - *PNAS*; Condamine et al. 2021 - *Nat. Comm.*; Weppe et al. 2021 - *PRSB*; Neubauer et al. 2022 - *PRSB*; Pino et al. 2022 - *Global Planet. Change*; Tarquini et al. 2022 - *Sci. Rep.*).

21. L185-187: Can the correlation values be compared like this directly, without considering the 95% HPDs in Table S2?

Yes, we can definitely compare the correlation parameters for a specific variable for which the effect may have changed through time. The MBD analyses apply a data scaling of the environmental variables to make such a comparison fair, which has been explained in the revised version of the manuscript (section *Methods*).

22. L187-191: Highly unclear sentence that feels like it belongs in the Discussion.

Moved.

23. L195-196: "Order-specific families"?

Changed to "specific families by orders".

24. L222: "It results"?

Changed to "resulted".

25. L224-225, L233-234: Feels like Discussion?

Deleted. This information already appeared in the discussion.

26. L238-240: To Discussion?

Deleted. This information already appeared in the discussion.

27. L240: "gymnosperm diversity"  "relative gymnosperm diversity"

Changed.

28. L242-244: This is pure Discussion, not Results. Also, because the additional factors are included in the analyses, they should be discussed.

Moved.

29. L247: groups  group

Changed.

30. L247-255: The hypothesized role of angiosperms for herbivore diversification should also be mentioned. There are numerous studies focusing on this.

Mentioned, despite it is more developed in the next paragraph.

31. L258: Unclear what "evolution of herbivory following cycles of host shifts" means.

Deleted "following cycles of host shifts" to be clearer because it was complementary information.

32. L283-284: This seems like a very strong assumption?

It is indeed. The first author is currently working on them.

33. L327-337: The end of the main Discussion focuses mainly on plant diversification, which was not the focus of the main analyses.

We disagree with this assumption. The cited paragraph is focusing in insect-plant relationship.

34. L348: "biases"

Changed.

35. L398-399: Odd sentence to end the Conclusions?

Modified.

36. L411-412: Unclear.

Rephrased.

37. L413: "PyRate has developed"?

The sentence has been rephrased.

38. L445-473: Since the emphasis of the study is on the effects of changes in relative diversity of different plant groups, it would seem more logical to first list plant-related variables and then the others.

Moved accordingly.

39. L467: If plant diversity is measured as relative numbers of genera, it should be mentioned here.

Done.

40. L481: Promoted?

We think it is correct to use this word here.

41. The relevant comment 8(viii) of Reviewer 2 has not been responded to, and x-axis units are still missing from Figs. 3 and 4.

We are not sure why there is confusion here because there is no unit scale for correlation parameters in the MBD analyses (see e.g. Lehtonen et al. 2017 - *Sci. Rep.*; Condamine et al. 2019 - *PNAS*; Condamine et al. 2021 - *Nat. Comm.*; Weppe et al. 2021 - *PRSB*; Neubauer et al. 2022 - *PRSB*; Pino et al. 2022 - *Global Planet. Change*; Jouault et al. 2022 – *Nat. Comm.* ; Tarquini et al. 2022 - *Sci. Rep.*).

Reviewers' Comments:

Reviewer #4:

Remarks to the Author:

The new version of the manuscript "The dual role of the angiosperm radiation in the global diversification of insects and insect pollinators" (NCOMMS-23-08227D) by Peris and Condamine is clearly further improved from the previous revision. As before, I think that the manuscript is very interesting and publishable. As detailed by Reviewer #3, fossil data and the PyRate approach used in the study have their limitations, but these are discussed in the Discussion, and the study in any case constitutes a significant step forward in the field.

I have only a few very minor comments, but I leave to the discretion of the editors if these are required or not:

1. As before, I think it is justifiable that plant diversity is measured at the genus level. However, it still remains unclear what is meant by "relative diversity." One would automatically assume that this means proportion of genera at a given time, but when looking at the relative diversities of angiosperms, gymnosperms, and spore plants given in the FigShare files, these do not add up to 1 per time period. Some clarification is clearly needed, especially if the use of proportional diversity creates correlations across the main plant taxa.

2. I would still refer to the comment by Reviewer 2 from the previous round that I referred to in my Comment 41. Reviewer 2 wrote: "8. If you keep the existing text and figures, I have minor suggested edits: viii) Figure 3 legend and axis labels: No units or title on the x --- Why not have the confidence intervals on the bars?"

I agree on the suggestion, and my formulation in comment 41 of the previous round regarding Figs. 3 and 4 was admittedly unclear. Instead of "units", I meant the figure should indicate that the X axes are correlations (at least below the lowermost plot) – now the reader has to read the figure legends to see what the plot axes represent. Also, the comment on cut bars added to the legend of Fig. 4 should appear in the legend of Fig. 3 as well.

3. I'm not entirely sure if the Editorial policy of Nat. Comm. is that data files used in analyses should be released at publication. If it is required, I would note that the data file package provided in the FigShare folder is not very transparent if someone would like to inspect or repeat the analyses, especially if the intention is to identify potential misassignments of taxa to particular groups.

RESPONSE TO REVIEWERS' COMMENTS

Reviewer #4 (Remarks to the Author):

The new version of the manuscript "The dual role of the angiosperm radiation in the global diversification of insects and insect pollinators" (NCOMMS-23-08227D) by Peris and Condamine is clearly further improved from the previous revision. As before, I think that the manuscript is very interesting and publishable. As detailed by Reviewer #3, fossil data and the PyRate approach used in the study have their limitations, but these are discussed in the Discussion, and the study in any case constitutes a significant step forward in the field.

Thank you for assessing again our revised study and point-by-point replies. We are delighted to read that you consider this new manuscript as improved and publishable. Thank you for the supportive comments.

I have only a few very minor comments, but I leave to the discretion of the editors if these are required or not:

1. As before, I think it is justifiable that plant diversity is measured at the genus level. However, it still remains unclear what is meant by "relative diversity." One would automatically assume that this means proportion of genera at a given time, but when looking at the relative diversities of angiosperms, gymnosperms, and spore plants given in the FigShare files, these do not add up to 1 per time period. Some clarification is clearly needed, especially if the use of proportional diversity creates correlations across the main plant taxa.

We are sorry for this misunderstanding. We are actually not sure why the relative diversity of plant groups do not sum up to 1 for the different groups of plants (angiosperms, gymnosperms, and spore plants). Angiosperm and gymnosperm diversities come from the study of Silvestro et al. (2015 - *New Phytol.*), and the diversity of spore plants come from an update of the Silvestro et al. (2015 - *New Phytol.*), published by Lehtonen et al. (2017 – *Sci. Rep.*). In this latter study, the authors have derived the relative diversities of all plant groups (available here: https://github.com/dsilvestro/PyRate/tree/master/example_files/predictors_MBDmodel). We have used these exact same files for angiosperms, gymnosperms, and spore plants. We agree that their diversities don't add up to 1 probably because they are not all from the same study. We have explained this point in the revised version.

2. I would still refer to the comment by Reviewer 2 from the previous round that I referred to in my Comment 41. Reviewer 2 wrote: "8. If you keep the existing text and figures, I have minor suggested edits: viii) Figure 3 legend and axis labels: No units or title on the x --- Why not have the confidence intervals on the bars?"

I agree on the suggestion, and my formulation in comment 41 of the previous round regarding Figs. 3 and 4 was admittedly unclear. Instead of "units", I meant the figure should indicate that the X axes are correlations (at least below the lowermost plot) – now the reader has to read the figure legends to see what the plot axes represent. Also, the comment on cut bars added to the legend of Fig. 4 should appear in the legend of Fig. 3 as well.

We respectfully disagree with the suggestion of stating that the plots are correlations in the same figures when both Figures 3 and 4 start their legends by: "Correlation trends...". This is the information that figure legends should contain, not necessarily explained in the same figure. Showing the credibility intervals will make the figures hard to read because of the high variation within each group and each variable. We have referred to Supplementary Data tables in which the mean/median and 95% HPD (credibility intervals) are clearly stated.

Following the second requirement of the reviewer, Figure 3 has been revised with the same information that is explained in the legend of Figure 4: "If any of the dates is out of their corresponding scale it is represented their value inside the box".

3. I'm not entirely sure if the Editorial policy of Nat. Comm. is that data files used in analyses should be released at publication. If it is required, I would note that the data file package provided in the FigShare folder is not very transparent if someone would like to inspect or repeat the analyses, especially if the intention is to identify potential misassignments of taxa to particular groups.

We agree with the reviewer and have presented data from FigShare folder in a cleaner and more organized way to ensure reproducibility including a ReadMe file.